

# Logarithmic negativity in out-of-equilibrium open free-fermion chains: An exactly solvable case

Vincenzo Alba[1*] and Federico Carollo[2]

**1** Dipartimento di Fisica dell' Università di Pisa and INFN, Sezione di Pisa, I-56127 Pisa, Italy
**2** Institut für Theoretische Physik, Universität Tübingen,
Auf der Morgenstelle 14, 72076 Tübingen, Germany

★ vincenzo.alba@unipi.it

## Abstract

We derive the quasiparticle picture for the fermionic logarithmic negativity in a tight-binding chain subject to gain and loss dissipation. We focus on the dynamics after the quantum quench from the fermionic Néel state. We consider the negativity between both adjacent and disjoint intervals embedded in an infinite chain. Our result holds in the standard hydrodynamic limit of large subsystems and long times, with their ratio fixed. Additionally, we consider the weakly-dissipative limit, in which the dissipation rates are inversely proportional to the size of the intervals. We show that the negativity is proportional to the number of entangled pairs of quasiparticles that are shared between the two intervals, as is the case for the mutual information. Crucially, in contrast with the unitary case, the negativity content of quasiparticles is not given by the Rényi entropy with Rényi index 1/2, and it is in general not easily related to thermodynamic quantities.

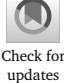

# 1 Introduction

Distinguishing genuine quantum correlations from statistical ones in quantum many-body systems is a daunting task. While for bipartite quantum systems in a pure state several *computable* quantum information motivated measures can be used to identify entanglement [1–4], this is more challenging for mixed-state systems. Open quantum systems undergoing dissipative Lindblad dynamics [5, 6] represent an important example of systems featuring mixed states. Recently, it has been shown that for one-dimensional free-fermion and free-boson systems it is possible to describe the dynamics of information-related quantities, such as von Neumann and Rényi entropies, as well as the mutual information, in the presence of generic quadratic dissipation [7–9]. This generalizes the well-known quasiparticle picture for entanglement spreading after quantum quenches in integrable systems [10–12]. Although the Rényi entropies and the mutual information are not proper measures of entanglement for mixed states [13], it has been shown in Ref. [8] that even for Lindblad dynamics the mutual information is sensitive to the presence of correlated pairs of quasiparticles. This is similar to closed quantum systems, although the dissipation dramatically affects the correlation content of the quasiparticles.

To make a step forward towards understanding entanglement dynamics in open quantum systems here we focus on the logarithmic negativity, which is a proper entanglement measure also for mixed states [14–18]. The computation of the logarithmic negativity is in general a challenging task. It can be computed effectively from the two-point correlation function only for free-boson systems [19]. For free fermionic ones it requires knowledge of the spectrum of the so-called partial transpose, which is not a Gaussian fermionic operator [20]. This means that the computational cost to extract the logarithmic negativity from a fermionic many-body wavefunction for a system with $L$ sites grows exponentially with $L$. Very recently, an alternative definition of negativity (that was dubbed *fermionic* negativity) has been proposed [21] for free fermions. This fermionic negativity can be computed from the two-point fermionic correlation functions. The computational cost to determine the fermionic logarithmic negativity

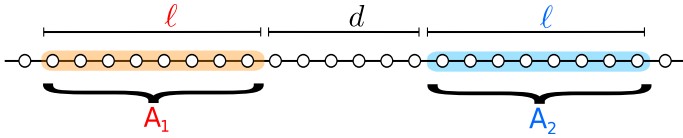

Figure 1: Logarithmic negativity $\mathcal{E}$ between two intervals $A_1$ and $A_2$ of equal length $\ell$ at distance $d$ embedded in an infinite chain.

grows only polynomially with $L$. For this reason here we restrict ourselves to the fermionic log-arithmic negativity. For generic interacting systems, both the standard and the fermionic loga-rithmic negativity can be computed with Matrix Product States (MPS) methods (see Ref. [22] for the standard negativity), at least at equilibrium. Importantly, both the standard and the fermionic negativity are proper entanglement measures for mixed states (see Ref. [21] for a careful comparison between them).

The negativity is attracting increasing attention as a tool to characterize universal aspects of equilibrium and out-of-equilibrium quantum many-body systems (see section 3). Interest-ingly, it has been shown in Ref. [23] that after quantum quenches in one-dimensional closed quantum integrable systems, both the standard negativity and the fermionic one become equal to half the Rényi mutual information with Rényi index 1/2. This has been verified for both free-fermion and free-boson models. For generic interacting systems it is quite challenging to build a quasiparticle picture to describe the full-time dynamics of Rényi entropies, although their value in the steady state can be determined [24–27]. Moreover, recent exact results for quenches in the so-called rule 54 chain [28], which is a "minimal model" for interacting inte-grable systems, suggest that Rényi entropies violate the quasiparticle picture paradigm. These results motivated a conjecture for the growth with time of the Rényi entropies in generic inter-acting integrable systems [29]. However, Ref. [30] showed that in the early-time regime and for contiguous subsystems the relation between Renyi-1/2 mutual information and logarith-mic negativity put forward in Ref. [23] still holds for any local quantum circuit, and therefore also for interacting integrable systems. Similar results were obtained in CFTs [31].

In the context of open quantum systems subject to a Lindblad dynamics the logarithmic negativity has not been explored much. Some numerical results were presented in Ref. [7], suggesting that in the presence of dissipation the negativity is not half of the Rényi mutual information with Rényi index 1/2, in contrast with closed systems [23]. Here we derive the quasiparticle picture for the fermionic logarithmic negativity after the quench from the fermionic Néel state in a tight-binding chain with homogeneous gain and loss of fermions. We consider the geometry sketched in Fig. 1, focusing on the entanglement between two intervals $A_1, A_2$ of length $\ell$ and placed at a distance $d$. The intervals are embedded in an infinite chain. Our results hold in the standard hydrodynamic limit of long times, large subsystem size, and large distances, i.e., $\ell, t, d \to \infty$, with the ratios $t/\ell$ and $d/\ell$ fixed and arbitrary. The loga-rithmic negativity decays expontially in time with a rate depending on the gain and loss rates $\gamma^{\pm}$. Since we consider times of order $\ell$, in order to observe a non-trivial time-evolution of the logarithmic negativity (and not an instantaneous convergence to its stationary value) we work in the weakly-dissipative hydrodynamic limit, obtained by taking vanishing $\gamma^{\pm} \to 0$, with fixed $\gamma^{\pm}\ell$. Our results show that the dynamics of the logarithmic negativity can be described within the framework of the quasiparticle picture. Specifically, we show that the logarithmic negativity, and hence the entanglement, is proportional to the number of entangled pairs of quasiparticles that are shared between the two intervals. Indeed, the structure of our formula for the logarithmic negativity is the same as that of the mutual information [8]. The contri-bution of the entangled quasiparticles to the negativity is time-dependent and it vanishes at long times, again, similar to the mutual information [8]. However, in contrast with the unitary case [23], this negativity content does not coincide with the Rényi entropy with Rényi index 1/2, and it is not, in general, straightforwardly related to known thermodynamic quantities.

The manuscript is organized as follows. In section 2 we introduce the tight-binding chain and discuss the treatment of the quench from the fermionic Néel state. In section 2.1 we review the Lindblad framework for gain and loss dissipation. In section 2.2 we discuss the quasiparti-cle picture for free systems with quadratic dissipation. In section 3 we introduce the fermionic logarithmic negativity. As a warm-up, we present in section 4 an *ab initio* derivation of the quasiparticle picture for the Rényi entropies [7]. This was obtained already in Ref. [7] using

the results of Ref. [32], although the alternative derivation that we present here is new and self-contained. In section 5 we derive the hydrodynamic picture for the logarithmic negativity. Specifically, in our approach this requires the calculation of the hydrodynamic behaviour of the moments of *ad hoc* modified fermionic correlation functions, which are obtained in section 5.2 and Appendix A. Our main result is discussed in section 5.3. In section 6 we benchmark our numerical results for the moments of the fermionic correlators (see section 6.1) and the logarithmic negativity (see section 6.2). We conclude in section 7. In Appendix A we provide some technical details about the results of section 5.2. In Appendix B we derive the formula for the negativity in terms of the fermionic correlation matrix for systems with fermion-number conservation.

## 2 Quantum quench in the open tight-binding chain

We consider the fermionic chain defined by the Hamiltonian

$$H = \frac{1}{2}\sum_{j=1}^{L}\left(c_j^\dagger c_{j+1} + c_{j+1}^\dagger c_j\right). \tag{1}$$

Here $c_j^\dagger$ and $c_j$ are canonical fermionic creation and annihilation operators with anticommutation relations $\{c_j^\dagger, c_l\} = \delta_{jl}$ and $\{c_j, c_l\} = 0$. For simplicity in (1) we assume that $L$ is an even integer. The Hamiltonian (1) is diagonalized by going to Fourier space by defining the fermionic operators $b_k := 1/\sqrt{L}\sum_j e^{ikj}c_j$, with the quasimomentum $k = 2\pi p/L$ and $p = 0, 1, \ldots, L-1$. The Hamiltonian (1) becomes diagonal as

$$H = \sum_k \varepsilon(k)b_k^\dagger b_k, \quad \text{with} \quad \varepsilon(k) := \cos(k), \tag{2}$$

where we defined the single-particle energy dispersion $\varepsilon(k)$. It is also convenient to define the group velocity of the fermionic excitations

$$v(k) := \varepsilon'(k) = \mathrm{d}\varepsilon(k)/\mathrm{d}k. \tag{3}$$

Here we focus on the nonequilibrium dynamics after the quench from the fermionic Néel state $|\mathrm{N}\rangle$ defined as

$$|\mathrm{N}\rangle := \prod_{j=1}^{L/2} c_{2j}^\dagger |0\rangle. \tag{4}$$

Specifically, at $t = 0$ the system is prepared in $|\mathrm{N}\rangle$. At $t > 0$ the chain undergoes unitary dynamics under the Hamiltonian (1). The fermionic correlation function $\widetilde{C}_{jl}$ is the central object to address entanglement related quantities in free-fermion systems [33]. This is defined as

$$\widetilde{C}_{jl} := \langle c_j^\dagger c_l\rangle = \langle\Psi(t)|c_j^\dagger c_l|\Psi(t)\rangle. \tag{5}$$

Under the closed-system dynamics implemented by $H$, the time-dependent correlation function $\widetilde{C}_{jl}(t)$ [cf. (5)] after the Néel quench is straightforwardly obtained as

$$\widetilde{C}_{jl} = \frac{1}{2}\delta_{jl} + \frac{1}{2}(-1)^l\int_{-\pi}^{\pi}\frac{dk}{2\pi}e^{ik(j-l)+2it\varepsilon(k)}. \tag{6}$$

In the following we consider the thermodynamic limit $L \to \infty$, as it is clear by the integration over the quasimomentum $k$. It is useful to exploit the invariance of the Néel state under

translation by two sites. Thus, we rewrite (6) as

$$\begin{pmatrix} \widetilde{C}_{2j,2l} & \widetilde{C}_{2j,2l-1} \\ \widetilde{C}_{2j-1,2l} & \widetilde{C}_{2j-1,2l-1} \end{pmatrix} = \int_{-\pi}^{\pi} \frac{dk}{2\pi} e^{2ik(j-l)} \hat{t}(k), \quad \text{with} \quad j, l \in [1, L/2]. \tag{7}$$

The factor 2 in the exponent in the integral reflects translation invariance by two sites. In (7) we have introduced the $2 \times 2$ matrix $\hat{t}(k)$ as

$$\hat{t}(k) = \frac{1}{2} \begin{pmatrix} 1 + e^{2it\varepsilon(k)} & -e^{2it\varepsilon(k)-ik} \\ e^{2it\varepsilon(k)+ik} & 1 - e^{2it\varepsilon(k)} \end{pmatrix}. \tag{8}$$

We can conveniently rewrite $\hat{t}(k)$ in terms of Pauli matrices as

$$\hat{t}(k) = \frac{1}{2} \left[ \mathbb{1}_2 + \sigma_{-i}^{(k)} e^{2it\varepsilon(k)} \right], \quad \text{with} \quad \sigma_{\pm i} := \sigma_z \pm i\sigma_y, \tag{9}$$

where we introduce the rotated Pauli matrices $\sigma_\alpha^{(k)}$ as

$$\sigma_\alpha^{(k)} := e^{-ik/2\sigma_z} \sigma_\alpha e^{ik/2\sigma_z}, \qquad \alpha = x, y, z, \tag{10}$$

with $\sigma_\alpha$ the standard Pauli matrices.

## 2.1 Lindblad evolution in the presence of gain and loss dissipation

In this work we study the out-of-equilibrium dynamics in the tight-binding chain (cf. (1)) with fermionic gain and loss processes. We employ the formalism of quantum master equations [5]. The Lindblad equation describes the evolution of the density matrix $\rho_t$ of the full system as

$$\frac{d\rho_t}{dt} = \mathcal{L}(\rho_t) := -i[H, \rho_t] + \sum_{j=1}^{L} \sum_{\alpha=\pm} \left( L_{j,\alpha} \rho_t L_{j,\alpha}^\dagger - \frac{1}{2} \left\{ L_{j,\alpha}^\dagger L_{j,\alpha}, \rho_t \right\} \right). \tag{11}$$

Here, $L_{j,\alpha}$ are the so-called Lindblad jump operators, which are defined as $L_{j,-} := \sqrt{\gamma^-} c_j$ and $L_{j,+} := \sqrt{\gamma^+} c_j^\dagger$, with $\gamma^\pm$ the gain and loss rates. Eq. (11) describes single-site incoherent absorption and emission of fermions which are homogeneous along the chain.

For free-fermion systems it is straightforward to obtain from (11) an equation for the fermionic two-point function $C_{jl} = \langle c_j^\dagger c_l \rangle = \text{Tr}(c_j^\dagger c_l \rho_t)$. The time-evolved matrix $C(t)$ is given by [8]

$$C(t) = e^{t\Lambda} C(0) e^{t\Lambda^\dagger} + \int_0^t dz \, e^{(t-z)\Lambda} \Gamma^+ e^{(t-z)\Lambda^\dagger}. \tag{12}$$

Here $\Lambda = ih - 1/2(\Gamma^+ + \Gamma^-)$, where $h$ encodes the effects of the Hamiltonian. Eq. (12) is obtained from (11) by using that $dC_{jl}(t)/dt = \text{Tr}(a_j^\dagger a_l \rho_t)$, with $\mathcal{L}$ defined in (11), and by applying Wick theorem (see Ref. [34] for further details). For the tight-binding chain considered here in (1) we have $h_{jl} = 1/2(\delta_{j+1,l} + \delta_{j,l+1})$. In (12), $\Gamma^\pm$ are $L \times L$ matrices describing gain and loss processes. Here we have $\Gamma_{jl}^\pm = \gamma^\pm \delta_{jl}$, reflecting that gain/loss dissipation acts separately on the different sites. The diagonal structure of the matrices $\Gamma_{jl}^\pm$ implies that $C_{jl}$ can be rewritten in terms of the correlation matrix $\widetilde{C}_{jl}(t)$ describing the quench in the absence of dissipation, i.e., with $\gamma^\pm = 0$. Precisely, one has

$$C = n_\infty (1-b)\mathbb{1} + b\widetilde{C}, \qquad b := e^{-(\gamma^+ + \gamma^-)t}, \qquad n_\infty := \frac{\gamma^+}{\gamma^+ + \gamma^-}. \tag{13}$$

Moreover, Eq. (13) suggests that it is convenient to modify the matrix $\hat{t}(k)$ (cf. (8)), introducing $\hat{t}'(k)$ as

$$\hat{t}'(k) = n_\infty(1-b)\mathbb{1}_2 + b\hat{t}(k) = \frac{1}{2}\begin{pmatrix} 2n_\infty(1-b) + b + be^{2it\varepsilon(k)} & -be^{2it\varepsilon(k)-ik} \\ be^{2it\varepsilon(k)+ik} & 2n_\infty(1-b) + b - be^{2it\varepsilon(k)} \end{pmatrix},$$
(14)

which can also be written as

$$\hat{t}'(k) = \frac{1}{2}\Big[ a\mathbb{1}_2 + b\sigma_{-i}^{(k)}e^{2it\varepsilon(k)} \Big], \quad \text{with} \quad a := 2n_\infty(1-b) + b,$$
(15)

where $\sigma_{-i}^{(k)}$ are defined in (9)-(10).

## 2.2 Quasiparticle picture for free systems with quadratic dissipation

Our goal is to determine the dynamics of the logarithmic negativity after the fermionic Néel quench in the tight-binding chain with gain and loss dissipation. Here we review the quasiparticle picture for free-fermion and free-boson systems in the presence of quadratic dissipation [8]. The quasiparticle picture for the entanglement dynamics [10–12, 35] can be generalized to describe the dynamics of quantum entropies, such as the von Neumann entropy and the Rényi entropies, and the mutual information [7–9] in the presence of generic *quadratic* dissipation [36]. Let us consider the Rényi entropies $S_A^{(n)}$ of a subsystem $A$ of length $\ell$ embedded in an infinite chain (see Fig. 1). The Rényi entropy of $A$ is given as [8]

$$S_A^{(n)}(t) = \ell \int_{-\pi}^{\pi} \frac{dk}{2\pi} \Big[ s_k^{(n),\text{YY}}(t) - s_k^{(n),\text{mix}}(t) \Big] \min(1, 2|v(k)|t/\ell) + \ell \int_{-\pi}^{\pi} \frac{dk}{2\pi} s_k^{(n),\text{mix}}(t),$$
(16)

where $v(k)$ (cf. (2) for the result in the tight-binding chain) is the fermion group velocity, which depends on the dispersion relation of the model. Crucially, Eq. (16) holds in the standard hydrodynamic limit $t, \ell \to \infty$ with their ratio fixed, which is the regime of validity for the standard quasiparticle picture for the entanglement spreading after quantum quenches [11, 12, 37]. In the presence of quadratic dissipation one has to take the weak-dissipation limit $\gamma \to 0$, with $\gamma\ell$ fixed, to ensure a nontrivial dynamics. Here $\gamma$ is the relevant dissipation rate, which measures the strength of the dissipative processes. For gain/loss dissipation this is the rate $\gamma^\pm$ (cf. (11)). The reason for taking the weakly-dissipative hydrodynamic limit is that at finite dissipation rate and for most dissipators, in the limit $t, \ell \to \infty$ and fixed $t/\ell$ one obtains a trivial scaling behaviour because the entropies and the mutual information would converge immediately to their stationary value, which for the mutual information is zero. In some cases, it is possible to apply Eq. (16) away from the weak-dissipation limit after redefining the group velocities $v(k)$ of the quasiparticles and rescaling by an exponential factor the entropies [38].

Let us now discuss the structure of (16). The first term has a similar structure as in the case without dissipation [11, 12]. In the absence of dissipation $s_k^{(n),\text{mix}} = 0$ and $s_k^{(n),\text{YY}}$ do not depend on time. Thus, Eq. (16) describes a linear growth of the entropies up to $t < \ell/(2v_{\text{max}})$, with $v_{\text{max}}$ the maximum velocity. At asymptotically long times $t \to \infty$ $S_A^{(n)}$ saturates to a volume-law behavior. Eq. (16) admits an interpretation in terms of entangled quasiparticle pairs [10]. After the quench, pairs of entangled quasiparticles are created uniformly in the systems. The quasiparticles travel as free particles. At the generic time $t$ the entanglement between $A$ and the rest is proportional to the number of shared entangled pairs, i.e., pairs that have one quasiparticle in $A$ and the other one in the complement of $A$. The entanglement content of the quasiparticles, i.e., their contribution to $S_A^{(n)}$ is given by the Yang-Yang entropies [37] $s_k^{(n),\text{YY}}$.

The Yang-Yang entropies are determined by the density $\rho_k$ of the Bogoliubov modes $b_k$ (cf. (2)) that diagonalize the model. The density is calculated over the pre-quench initial state

$|\psi_0\rangle$. Specifically, we have

$$s_k^{(n),\text{YY}} := \frac{1}{1-n} \ln\left(\rho_k^n + (1-\rho_k)^n\right), \quad \text{with} \quad \rho_k := \langle\psi_0|b_k^\dagger b_k|\psi_0\rangle. \tag{17}$$

Here $s_k^{(n),\text{YY}}$ is the density of Rényi entropy of the Generalized Gibbs Ensemble [39–44] (GGE) that describes local properties of the steady state after the quench.

This scenario changes dramatically in the presence of quadratic dissipation [8]. First, the new term $s_k^{(n),\text{mix}}$ appears. This is purely dissipative and it can be obtained as the density of Rényi entropies of the full system. Indeed, the first term in (16) cannot contribute to the entropies of the full system because it describes the contribution of correlated pairs that are shared between $A$ and its complement. If $A$ is the full system both members of a correlated pair are within $A$ and hence they cannot contribute. This means that only the second term in (16) contributes to the full-system entropy. Clearly, in the unitary case the full system is in a pure state at any time and $s_k^{(n),\text{mix}} = 0$. For free-fermion and free-boson models $s_k^{(n),\text{mix}}$ is straightforwardly extracted from the two-point correlation function in momentum space [8,9]. As it is clear from (16), dissipation affects the correlation between the quasiparticles as well. First, the same $s_k^{(n),\text{mix}}$ appears in the first term in (16). The minus sign reflects that dissipation diminishes the correlation of the pairs. Moreover, although $s_k^{(n),\text{YY}}$ in (16) has the form of a Yang-Yang entropy (cf. (17)), the density $\rho_k$ from which it is obtained is no longer that of the original modes $b_k$. It has been conjectured in Ref. [8] that in the presence of dissipation the density $\rho_k$ to be used in (17) is that of the eigenmodes $\beta_k$ of the map $\mathcal{L}^*$, which is the dual map — the one acting on observables — of the generator $\mathcal{L}$ appearing in Eq. (11). In the weak-dissipation limit $\beta_k$ generically satisfy [8]

$$\mathcal{L}^*(\beta_k) = -\left(\frac{\gamma_k}{2} + i\varepsilon(k)\right)\beta_k, \tag{18}$$

where $\varepsilon(k)$ is the dispersion of the model without dissipation, and $\gamma_k$ are dissipation rates that are easily calculable. Both $\varepsilon(k)$ and $\gamma_k$ are real. For the case of free fermion with gain and losses, Eq. (18) simplifies because $\beta_k$ coincides with the Fourier transformed fermionic operators $c_k$. Indeed, one can easily check that Eq. (18) is satisfied with $\gamma_k = \gamma^+ + \gamma^-$ and $\varepsilon_k = \cos(k)$. The first term originates from the dissipative part (cf. second term in (11)), whereas the second one from the unitary part of the Liouvillian (first term in in (11)). Notice that for gain and loss dissipation Eq. (18) holds also at generic $\gamma^\pm$, i.e., away from the weak-dissipation limit. However, Eq. (18) is expected to be valid in general in the weak-dissipation limit because the eigenvectors and eigenvalues of $\mathcal{L}^*$ should be linear in the dissipation rates, in the limit of weak dissipation. Moreover, in the absence of dissipation, $\beta_k$ become the fermionic operators that diagonalize the system, and the eigenvalue of $\mathcal{L}^*$ becomes $-i\varepsilon(k)$.

Now, $s_k^{(n),\text{YY}}$ in (16) is the Yang-Yang entropy (17) calculated from the density $\rho_k$ of the modes $\beta_k$, i.e., $\rho_k = \langle\psi_0|\beta_k^\dagger\beta_k|\psi_0\rangle$. In contrast with the density of the modes $b_k$, which in the unitary case is time-independent, the density of $\beta_k$ is time-dependent, implying that $s_k^{(n),\text{YY}}$ depends on time. By computing $\mathcal{L}^*(\beta_k^\dagger\beta_k)$ and using also (18), the evolution of $\rho_k = \langle\psi_0|\beta_k^\dagger\beta_k|\psi_0\rangle$ in the weakly-dissipative limit is obtained as [8]

$$\rho_k(t) = e^{-\gamma_k t}\rho_k(0) + \frac{\alpha_k}{\gamma_k}(1 - e^{-\gamma_k t}). \tag{19}$$

Here $\alpha_k$ is, again, a function that depends on the dissipation and that can be easily calculated for generic free systems with quadratic dissipation [8,9]. Eq. (16) was derived *ab initio* for a quench in the Kitaev chain with arbitrary quadratic dissipation in Ref. [9].

For the case of diagonal gain and loss dissipation, the decay rates $\gamma_k$ in (18) do not depend on $k$, and one has that $\gamma_k = \gamma^+ + \gamma^-$, and $\alpha_k = \gamma^+$ (cf. (19)). It is also important to stress that

while for generic dissipation the modes $\beta_k$ are different from the original Bogoliubov modes $b_k$ (see, for instance, Ref. [9]), for several types of dissipation one has that $\beta_k = b_k$. The gain/loss dissipation that we treat here provides one of the simplest examples for which this happens.

## 3  Fermionic logarithmic negativity: Definitions

Here we are interested in the entanglement between two non complementary regions $A_1$ and $A_2$ (see Fig. 1). Let us first introduce the Rényi entropies $S_W^{(n)}$ of a subsystem $W = A_1, A_2, A_1 \cup A_2$, which are defined as

$$S_W^{(n)} := \frac{1}{1-n} \ln \left( \mathrm{Tr} \rho_W^n \right). \tag{20}$$

Here we often consider the von Neumann entropy, which corresponds to the limit $n \to 1$ [1–4].

Crucially, since the interval $A = A_1 \cup A_2$ in Fig. 1 is in general in a mixed state, neither the von Neumann nor the Rényi entropies can be used to quantify the entanglement between $A_1$ and $A_2$ [1–4]. Instead, one can use the logarithmic negativity [14–18] $\mathcal{E}$, which is a computable entanglement measure for mixed states. To define $\mathcal{E}$ one has to introduce the partially-transposed density matrix $\rho_A^{T_2}$. The partial transposition is taken with respect to one of the intervals (here $A_2$). $\rho_A^{T_2}$ is defined as

$$\left\langle e_i^{(1)}, e_j^{(2)} | \rho_A^{T_2} | e_k^{(1)}, e_l^{(2)} \right\rangle = \left\langle e_i^{(1)}, e_l^{(2)} | \rho_A | e_k^{(1)}, e_j^{(2)} \right\rangle, \tag{21}$$

with $e_i^{(1)}, e_j^{(2)}$ two bases for $A_1$ and $A_2$, respectively. The partial transpose is not positive-definite, and its negative eigenvalues quantify the entanglement between the two intervals. The logarithmic negativity is defined as

$$\mathcal{E} = \ln \left( \mathrm{Tr} | \rho_A^{T_2} | \right). \tag{22}$$

For free-boson systems the negativity can be computed from the two-point correlation function [19]. For free-fermion systems the partially transposed reduced density matrix can be decomposed as [20]

$$\rho_A^{T_2} = e^{-i\pi/4} O_+ + e^{i\pi/4} O_-, \tag{23}$$

where $O_\pm$ are gaussian operators. Crucially, while the spectrum of $O_\pm$ can be effectively computed from that of the fermionic two-point function, this cannot be done for $\rho_A^{T_2}$. As a consequence, the negativity cannot be easily calculated, not even for free-fermion models, although several results have been obtained in the literature [45–47] for the moments of the partial transpose. Recently, it has been shown that starting from the decomposition (23), it is possible to construct an alternative measure of entanglement for mixed-state systems. This has been dubbed fermionic negativity [21, 48–50]. The fermionic negativity is defined as

$$\mathcal{E} := \ln \mathrm{Tr} \sqrt{O_+ O_-}. \tag{24}$$

Notice that here we use the same symbol $\mathcal{E}$ for both the standard negativity (cf. (22)) and the fermionic one. In the following sections we will always refer to the fermionic negativity.

Since the product $O_+ O_-$ in (24) is a gaussian operator because $O_\pm$ is gaussian, the fermionic negativity (24) can be computed effectively in terms of fermionic two-point functions. The central object is the fermionic correlation matrix $C_{jl}$ (cf. (12)). Let us define the matrix $G_{jl}$ as

$$G_{jl} := 2C_{jl} - \delta_{jl}. \tag{25}$$

We now consider the partition in Fig. 1. We focus on two intervals $A_1$ and $A_2$ of equal length $\ell$ at distance $d$. We now define $G_{jl}^{\alpha\beta}$ with $\alpha, \beta = 1, 2$ as the restricted correlator with $j \in A_\alpha$ and $l \in A_\beta$. The matrix $G_A$ is rewritten as

$$G_A = \begin{pmatrix} G^{11} & G^{12} \\ G^{21} & G^{22} \end{pmatrix}, \tag{26}$$

where $G^{\alpha\beta}$ are $\ell \times \ell$ matrices. We now define the matrices $G_A^\pm$ as

$$G_A^\pm = \begin{pmatrix} G^{11} & \pm i G^{12} \\ \pm i G^{21} & -G^{22} \end{pmatrix}. \tag{27}$$

These are the covariance matrices of the operators $O^\pm$ introduced in (23). Finally, the negativity is a function of the spectrum of $C_A$, which is the fermionic correlator $C$ restricted to $A$, and that of $G_A^{\mathrm{T}}$, which is defined as

$$G_A^{\mathrm{T}} := \frac{1}{2}\Big[\mathbb{1}_{2\ell} - (\mathbb{1}_{2\ell} + G_A^+ G_A^-)^{-1}(G_A^+ + G_A^-)\Big]. \tag{28}$$

Here $\mathbb{1}_{2\ell}$ is the $2\ell \times 2\ell$ identity matrix. $G_A^{\mathrm{T}}$ is the covariance matrix of the product $O^+ O^-/\mathrm{Tr}(O^+ O^-)$. The negativity is defined as [51]

$$\mathcal{E} := \sum_{j=1}^{2\ell} \ln\Big[\mu_j^{1/2} + (1-\mu_j)^{1/2}\Big] + \sum_{j=1}^{2\ell} \frac{1}{2} \ln\Big[\lambda_j^2 + (1-\lambda_j)^2\Big], \tag{29}$$

where $\mu_j$ are the eigenvalues of $G_A^{\mathrm{T}}$ and $\lambda_j$ of $C_A$. The second term in (29) originates from the normalization $\mathrm{Tr}(O^+ O^-) = \mathrm{Tr}\rho_A^2$ (see, for instance, Ref. [52]). Importantly, Eq. (29) holds for free-fermion systems that preserve the fermion number. For generic free-fermion systems a generalization of (29) exists in terms of the correlation function of Majorana fermions [21,50]. In the presence of gain/loss dissipation the fermion number is not preserved, although at any time one has $\langle c_j^\dagger c_l^\dagger \rangle = \langle c_j c_l \rangle = 0$. In Appendix B we show that this condition is sufficient to ensure the validity of (29).

The logarithmic negativity has been employed to characterize entanglement in systems of harmonic oscillators [19,53–58], spin models [22,59–70], Conformal Field Theory (CFT) [18, 71–77]. The out-of-equilibrium dynamics after quantum quenches has received a lot of attention [23,31,78–84]. In particular, it has been shown in Ref. [23] that for large intervals, long times, and large distance (see Fig. 1) $\ell, t, d \to \infty$ with the ratios $\ell/t$ and $d/t$ fixed, the standard negativity and the fermionic one become

$$\mathcal{E} = \frac{1}{2} I_{A_1:A_2}^{(1/2)}, \tag{30}$$

where $I_{A_1:A_2}^{(1/2)}$ is the Rényi mutual information with Rényi index $1/2$. Eq. (30) was proposed in Ref. [23], and it was verified for quantum quenches in free-boson and free-fermion systems. It is natural to expect that Eq. (30) holds for generic interacting integrable systems. Very recently, it has been shown that in the early-time regime Eq. (30) holds true in quenches with generic local quantum circuits [30]. Eq. (30) is intriguing because in general the mutual information between $A_1$ and $A_2$ is not a good measure of their entanglement but only an upper bound [85].

## 4 Warm-up: Quantum entropies in the presence of gain and loss dissipation

As a warm-up, before deriving the quasiparticle picture for the fermionic negativity, here we provide an alternative direct derivation of the results of Ref. [7]. In contrast with Ref. [7] the

derivation that we present here is *ab initio*, although we rely on the same analytic techniques employed in Ref. [86]. In section 4.1 we derive the behavior of the moments of the correlation matrix $\text{Tr}(C_A^n)$. In section 4.2 we discuss the Rényi entropies.

## 4.1 Moments of the correlators $\text{Tr}(C_A^n)$

Here we determine the scaling of $\text{Tr}(C_A^n)$ in the hydrodynamic limit $t, \ell \to \infty$, with their ratio $t/\ell$ fixed. At the end of the derivation we will also discuss the weakly-dissipative limit by taking $\gamma^{\pm} \to 0$ with the product $\gamma^{\pm}\ell$ fixed. To derive our main results we employ the approach of Ref. [86]. The correlator $C_A$ is defined in (13). We can use the trivial identity

$$\sum_{z=1}^{\ell/2} e^{2izk} = \frac{\ell}{4} \int_{-1}^{1} d\xi\, w([k]_\pi) e^{i(\ell\xi+\ell+2)[k]_\pi/2}, \quad \text{with} \quad w(k) := \frac{k}{\sin(k)}, \tag{31}$$

where we introduced the notation $[x]_\pi = x \bmod \pi$. The mod $\pi$ reflects the factor 2 in the exponent in the left hand side in (31), and it is due to the fact that the initial state is not invariant under one-site translation, although it is invariant under two-site translations. Notice that (31) is different from a similar identity used in Ref. [86], which deals with one-site translation invariant initial states. Eq. (31) allows us to write

$$\text{Tr}(C_A^n) = \left(\frac{\ell}{4}\right)^n \int_{[-\pi,\pi]^n} \frac{d^n k}{(2\pi)^n} \int_{-1}^{1} d^n\xi\, D(\{k\})F(\{k\}) e^{i\ell \sum_{j=0}^{n-1} \xi_{j+1}([k_{j+1}-k_j]_\pi)/2}. \tag{32}$$

Here we defined

$$D(\{k\}) = \prod_{j=0}^{n-1} w\left([k_j - k_{j-1}]_\pi\right), \tag{33}$$

$$F(\{k\}) = \text{Tr} \prod_{j=0}^{n-1} \hat{t}'\left(k_j\right), \tag{34}$$

where $t'(k)$ is defined in (15). In deriving (32) from (31), we neglect the factor $\ell+2$ because it contributes with a phase. Notice that the quasimomenta in $F$ (cf. (34)) are not defined mod $\pi$. It is convenient to define new variables $\zeta_j$ as

$$\zeta_0 = \xi_1, \tag{35}$$

$$\zeta_i = \xi_{i+1} - \xi_i, \quad i \in [1, n-1]. \tag{36}$$

This allows us to write (32) as

$$\text{Tr}(C_A^n) = \left(\frac{\ell}{4}\right)^n \int_{[-\pi,\pi]^n} \frac{d^n k}{(2\pi)^n} \int_{R_\xi} d^n\zeta_i\, D(\{k\})F(\{k\}) e^{-i\ell \sum_{j=0}^{n-1} \sum_{l=0}^{j} \zeta_l([k_{j+1}-k_j]_\pi)/2}. \tag{37}$$

Here the integration domain $R_\xi$ for $\zeta_i$ is

$$R_\xi: -1 \le \sum_{j=0}^{p-1} \zeta_j \le 1, \qquad p \in [1, n]. \tag{38}$$

The strategy to determine the behaviour of (37) in the space-time scaling limit is to use the stationary phase approximation for the integrals over $k_1, \ldots, k_{n-1}$ and $\zeta_1, \ldots, \zeta_{n-1}$. It is easy to check that stationarity with respect to the variables $\zeta_1, \ldots, \zeta_{n-1}$ implies that

$$[k_{j+1} - k_j]_\pi = 0, \qquad \forall\, j \in [0, n-1]. \tag{39}$$

This also implies that the integrand in (37) does not depend on $\zeta_0$. Thus, the integration over $\zeta_0$ is trivial and we obtain

$$
\text{Tr}(C_A^n) = \left(\frac{\ell}{4}\right)^n \int\limits_{[-\pi,\pi]^n} \frac{d^n k}{(2\pi)^n} \int d^{n-1}\zeta_i D(\{k\})F(\{k\})e^{-i\ell \sum_{j=0}^{n-1}\sum_{l=1}^{j}\zeta_l([k_{j+1}-k_j]_\pi)/2}\mu\left(\{\zeta_j\}\right),
$$
(40)

where we introduced the integration measure $\mu$ as

$$
\mu(\{\zeta_j\}) = \max\left[0, \min_{j\in[0,n-1]}\left[1-\sum_{k=1}^{j}\zeta_k\right] + \min_{j\in[0,n-1]}\left[1+\sum_{k=1}^{j}\zeta_k\right]\right].
$$
(41)

Now we can replace $k_j \to k_0$ in $D(\{k_j\})$ because it depends only on $[k_j - k_{j-1}]_\pi$ to obtain

$$
D(\{k_j\}) \to 1.
$$
(42)

The function $F(\{k_j\})$ requires some care. We can expand it as

$$
F = \frac{1}{2^n}\sum_{p=0}^{n-1}\sum_{j_1<\cdots<j_p=1}^{n-1}\text{Tr}\left[\left(a\mathbb{1}_2 + b\sigma_{-i}^{(k_0)}e^{2it\varepsilon(k_0)}\right)b^p\sigma_{-i}^{(k_{j_1})}\sigma_{-i}^{(k_{j_2})}\cdots\sigma_{-i}^{(k_{j_p})}\right]e^{2it\sum_{l=1}^{p}\varepsilon(k_{j_l})},
$$
(43)

where $a$, $b$ and $\sigma_{-i}$ are defined in (15), and we isolated the term with $k_0$. It is worth noticing that in Eq. (43) the time-dependent term appearing in the exponent is a number. This is specific of the tight-binding chain. For generic quenches in the $XY$ chain the term in the exponent is a matrix [86].

To perform the trace in (43) we observe that $\sigma_{-i}^{(k)}\sigma_{-i}^{(k')} = 0$ if $k \to k'$. On the other hand, one has that that $\text{Tr}(\sigma_{-i}^{(k_0)}\sigma_{-i}^{(k_{j_1})}\ldots\sigma_{-i}^{(k_{j_{p-1}})}) = 2^{p+1}$ if from the stationary solution (39) one selects the alternating pattern as $k_{j_1} = k_0 + \pi, k_{j_2} = k_0,\ldots$, and it vanishes otherwise. For the second term, one has $\text{Tr}(\sigma_{-i}^{(k_{j_1})}\sigma_{-i}^{(k_{j_2})}\ldots\sigma_{-i}^{(k_{j_p})}) = 2^{p+1}$ for both the alternating patterns $k_{j_1} = k_0, k_{j_2} = k_0 + \pi,\ldots$ and $k_{j_1} = k_0 + \pi, k_{j_2} = k_0,\ldots$. This implies that the first term in the trace in (43) gives nonzero contribution only for even $p$, whereas the second term contributes to odd $p$. Thus, we can rewrite (43) as

$$
\begin{aligned}
F(\{k_j\}) \propto{} & 2^{-n}\sum_{p=0}^{\lfloor(n-1)/2\rfloor}\binom{n-1}{2p}a^{n-2p}(2b)^{2p}e^{2it\sum_{l=1}^{2p}\varepsilon(k_l)} \\
& + 2^{-n}\sum_{p=0}^{\lfloor(n-2)/2\rfloor}\binom{n-1}{2p+1}a^{n-2p-2}(2b)^{2p+2}e^{2it\sum_{l=0}^{2p+1}\varepsilon(k_l)}.
\end{aligned}
$$
(44)

Here we used the invariance under relabelling of the momenta $k_{j_l}$ to replace $k_{j_l} \to k_l$ in the phase factor. However, we are not allowed to replace $k_l$ with their stationary values before performing the stationary phase approximation. The binomials in (44) are the number of terms containing $2p$ and $2p+1$ quasimomenta, and that are the same under exchange of the momentum label. The factors $2^{2p}$ and $2^{2p+2}$ are the results of the trace operation. Finally, the proportionality symbol $\propto$ in (44) is because there is an extra constant that originates from the total number of stationary points in (39). Indeed, in principle Eq. (39) corresponds to $2^n$ stationary points. This proliferation of the stationary points is due to the invariance under shift by $\pi$ (cf. (39)), which reflects translation invariance by two sites. Again, this is different from Ref. [86]. However, the presence of the strings of $\sigma_{-i}^{(k_j)}$ selects one of the two patterns $k_1 = k_0, k_2 = k_0 + \pi,\ldots$ or $k_1 = k_0 + \pi, k_2 = k_0,\ldots$. For the first term in (44) there is a

remaining overall factor $2^{n-1-2p}$ that originates from the $n-1-2p$ quasimomenta that do not appear in the string of $\sigma_{-i}$. Moreover, there is an extra factor 2 because both alternating patterns $\{\bar{k}_1, \bar{k}_2, \ldots, \bar{k}_{2p}\} = \{k_0, k_0 + \pi, k_0, \ldots\}$ and $\{\bar{k}_1, \bar{k}_2, \ldots, \bar{k}_{2p}\} = \{k_0 + \pi, k_0, k_0 + \pi, \ldots\}$ contribute. This is different for the second term in (44). The stationary phase treatment of the quasimomenta that do not appear in the phase factor gives a factor $2^{n-2-2p}$. There is no extra factor 2 because only the quasimomenta pattern $\{\bar{k}_1, \bar{k}_2, \bar{k}_3, \ldots\} = \{k_0 + \pi, k_0, k_0 + \pi, \ldots\}$ gives a nonzero contribution after taking the trace in (43). By putting together all the factors, the result is that one can drop the prefactors $2^{-n}, 2^{2p}$ and $2^{2p+2}$ in (44) to obtain

$$F = \sum_{p=0}^{\lfloor (n-1)/2 \rfloor} \binom{n-1}{2p} a^{n-2p} b^{2p} e^{2it \sum_{l=1}^{2p} \varepsilon(k_l)} + \sum_{p=0}^{\lfloor n/2 \rfloor - 1} \binom{n-1}{2p+1} a^{n-2p-2} b^{2p+2} e^{2it \sum_{l=0}^{2p+1} \varepsilon(k_l)}. \tag{45}$$

The strategy is now to apply the stationary phase approximation to the integral in the $2n-2$ variables $k_1, k_2, \ldots, k_{n-1}, \zeta_1, \zeta_2, \ldots, \zeta_{n-1}$. For the first term in (45) with the quasimomenta $k_1, k_2, \ldots, k_{2p}$ appearing in the phase factor one obtains the stationary points $\bar{k}_j$ and $\bar{\zeta}_j$ as

$$\{\bar{k}_1, \bar{k}_2, \ldots, \bar{k}_{2p}\} = \{k_0, k_0 + \pi, \ldots k_0 + \pi\} \cup \{k_0 + \pi, k_0, \ldots k_0\}, \tag{46}$$

$$\bar{\zeta}_l = \pm 4 \frac{t}{\ell} (-1)^l \varepsilon'(k_0), \qquad l = 1, \ldots, 2p, \tag{47}$$

$$\bar{\zeta}_j = 0, \qquad l > 2p. \tag{48}$$

Here we have to choose only one of the patterns in (46) because they give the same result, and this was already taken into account in (45). The sign of $\bar{\zeta}_j$ in (47) is different for the two patterns in (46). However, this sign does not affect the final result. The reason is that $\bar{\zeta}_l$ enter only in the function $\mu(\{\zeta_j\})$ (cf. (41)), which remains the same under change of the sign of $\bar{\zeta}_l$. The stationary phase treatment of the second term in (45) is similar. The result is that only the second pattern in (46) contributes. We now use the formula for the stationary phase approximation [87]

$$\int_{\mathcal{D}} d^N x\, p(\boldsymbol{x}) e^{i\ell q(\boldsymbol{x})} \to \left(\frac{2\pi}{\ell}\right)^{N/2} p(\boldsymbol{x}_0) |\det H|^{-1/2} \exp\left[i\ell q(\boldsymbol{x}_0) + i\pi \frac{\sigma_A}{4}\right]. \tag{49}$$

Here $p(\boldsymbol{x})$ and $q(\boldsymbol{x})$ are functions, $\mathcal{D}$ denotes the domain of integration and $\ell$ is the large parameter. In (49), we denote by $\boldsymbol{x}_0$ the stationary point that is solution of $\boldsymbol{\nabla} q(\boldsymbol{x}_0) = 0$. The Hessian matrix $H$ is given by $H = \partial_{x_i} \partial_{x_j} q(\boldsymbol{x})$. The signature $\sigma$ of the Hessian is calculated as the difference between the number of positive and negative eigenvalues of $H$, and in our case it is zero. Moreover, in our case $|\det H|^{-1/2} = 2^{n-1}$, and the phase in (49) vanishes.

In using (49), we observe that the term with $p = 0$ in the first sum in (45) gives (cf. (41)) $\mu(\{\zeta_j\}) = 2$. On the other hand, all the other terms give the same result as

$$\mu(\{\zeta_j\}) = 2 \max(0, 1 - 2t/\ell |v(k_0)|), \tag{50}$$

with $v(k_0)$ the group velocity (cf. (3)). Finally, it is clear from (45) that the term with $p = 0$ contributes with $a^n$, whereas the remaining sum gives $((a+b)^n + (a-b)^n - 2a^n)/2$. This implies that

$$\text{Tr}(C_A^n) = \frac{a^n}{2^n} \ell + \frac{(a-b)^n + (a+b)^n - 2a^n}{2^{n+1}} \int_{-\pi}^{\pi} \frac{dk}{2\pi} \max(0, \ell - 2t|v(k)|), \tag{51}$$

where we replaced $k_0 \to k$. This can be rewritten as

$$\text{Tr}(C_A^n) = \ell \int_{-\pi}^{\pi} \frac{dk}{2\pi} \left[\left(\frac{a}{2}\right)^n - \frac{(a+b)^n + (a-b)^n}{2^{n+1}}\right] \min(1, 2|v(k)|t/\ell) + \ell \int_{-\pi}^{\pi} \frac{dk}{2\pi} \frac{(a+b)^n + (a-b)^n}{2^{n+1}}. \tag{52}$$

Now, the first term in (52) admits a quasiparticle interpretation. Indeed, the function $\min(1, 2|v(k)|t/\ell)$ is the number of pairs of entangled quasiparticles with quasimomenta $k$ and $-k$ that are shared between $A$ and the rest at time $t$. The second term is proportional to the volume of $A$ and it is not due to the pairs of quasiparticles.

## 4.2 Rényi entropies and von Neumann entropy

Eq. (52) allows us to obtain the behaviour of $\mathrm{Tr}(\mathcal{F}(C_A))$, where $\mathcal{F}(z)$ is an arbitrary function that admits a Taylor expansion in $z = 0$. After expanding $\mathcal{F}(C_A)$ and using (52), one obtains

$$\mathrm{Tr}(\mathcal{F}(C_A)) = \ell \int_{-\pi}^{\pi} \frac{dk}{2\pi} \left[ \mathcal{F}(a/2) - \frac{\mathcal{F}((a+b)/2) + \mathcal{F}((a-b)/2)}{2} \right] \min(1, 2|v(k)|t/\ell)$$
$$+ \ell \int_{-\pi}^{\pi} \frac{dk}{2\pi} \frac{\mathcal{F}((a+b)/2) + \mathcal{F}((a-b)/2)}{2}. \quad (53)$$

The functions $\mathcal{F}_n(x)$ and $\mathcal{F}_1(x)$ that correspond to the Rényi entropies $S_A^{(n)}$ and the von Neumann entropy $S_A$ read

$$\mathcal{F}_n(x) := \frac{1}{1-n} \ln(x^n + (1-x)^n), \quad (54)$$
$$\mathcal{F}_1(x) := -x \ln(x) - (1-x) \ln(1-x). \quad (55)$$

After using (54) and (55) in (53) one recovers the results of Ref. [7]. Let us also discuss the non dissipative limit $\gamma^{\pm} \to 0$. In that limit $a, b \to 1$ (see (13) and (15)), and one obtains

$$\mathrm{Tr}(\mathcal{F}_n(C_A)) = \int_{-\pi}^{\pi} \frac{dk}{2\pi} \left\{ \left[ \mathcal{F}_n(1/2) - \frac{\mathcal{F}_n(0) + \mathcal{F}_n(1)}{2} \right] \min(\ell, 2|v(k)|t) + \frac{\ell}{2} (\mathcal{F}_n(0) + \mathcal{F}_n(1)) \right\}. \quad (56)$$

Since $\mathcal{F}_n(0) = \mathcal{F}_n(1) = 0$ for any $n$, only the first term in the square brackets in (56) survives. One has that $\mathcal{F}_n(1/2) = \ln(2)$ for any $n$, which implies that for the Néel quench all the Rényi entropies are equal. Moreover, the density of Rényi entropy does not depend on $k$. Both these two features are specific for the Néel quench. Finally, we should observe that Eq. (54) holds in the limit $\ell, t \to \infty$ with the ratio $t/\ell$ finite. In particular, for this simple case of diagonal dissipation Eq. (54) holds also at finite $\gamma^{\pm}$. Still, in the limit $t \to \infty$ one has that $S_A^{(n)} \to 0$ for any $n$. To have a nontrivial dynamics we take the weakly-dissipative hydrodynamic limit $t, \ell \to \infty$, $\gamma^{\pm} \to 0$, with $t/\ell$ and $\gamma^{\pm}\ell$ fixed [7–9].

# 5 Fermionic logarithmic negativity

We are now ready to derive the behaviour of the fermionic logarithmic negativity in the weakly-dissipative hydrodynamic limit. The strategy is as follows. We first focus on the moments $\mathrm{Tr}[(G^+ G^-)^n]$. In the following section we drop the subscript $A$ in $G_A^{\pm}$, as it is clear that we will always consider the correlators restricted to subsystem $A$. The results are presented in section 5.1. Then we determine the hydrodynamic behaviour of some modified moments of $G^+ G^-$. These are obtained by expanding the $n$-th power of $G^+ G^-$ and inserting an arbitrary number $m$ of "misplaced" $G^{\pm}$. These insertions break the alternating pattern of $G^+ G^-$, creating "defects" at places where the same operator is present on consecutive positions. The hydrodynamic limit of these defective moments is derived in section 5.2 The derivation is reported in Appendix A. Finally, in section 5.3 we provide the result for the fermionic logarithmic negativity.

## 5.1 Moments $\text{Tr}[(G^+G^-)^n]$

Let us now consider the moments of the product $G^+G^-$ (cf. (27)), i.e, $\text{Tr}[(G^+G^-)^n]$ for arbitrary $n$. Here we focus on a subsystem $A$ of length $2\ell$, which is further divided into two adjacent equal-length intervals $A_1$ and $A_2$ (see Fig. 1). Let us start by defining the Fourier transform $\hat{t}''(k)$ of the matrix (cf. (26)) $G = \mathbb{1} - 2C$ as

$$\hat{t}''(k) = a'\mathbb{1}_2 - b\sigma_{-i}^{(k)}e^{2it\varepsilon(k)}, \qquad a' := 1 - a. \tag{57}$$

The matrices $G^\pm$ (cf. (27)) are written as

$$G^\pm = \int_{-\pi}^{\pi} \frac{dk}{2\pi} e^{2ik(j-l)}\sigma_{\mp}^{(k\ell)} \otimes \left[a'\mathbb{1}_2 - b\sigma_{-i}^{(k)}e^{2it\varepsilon}\right], \quad j,l = 1,\dots,\ell. \tag{58}$$

In (58) we defined $\sigma_{\pm}^{(k\ell)}$ as

$$\sigma_{\pm}^{(k\ell)} := \sigma_z^{(k\ell)} \pm \sigma_y^{(k\ell)}, \tag{59}$$

where $\sigma_\alpha^{(k\ell)}$ are the rotated Pauli matrices defined in (10). Notice that $\ell$ appears in the rotation angle in (59). This is important when using the stationary phase approximation. The tensor product with $\sigma_{\mp}^{(k\ell)}$ in (58) accounts for the fact that the indices $j,l$ are shifted by $\ell$ when considering the blocks $G^{12}$ and $G^{21}$ (cf. (26)). The structure in (58) is straightforwardly generalizable to quenches from other initial states by changing the term in the square brackets. Similar to (32), one can write the moments $\text{Tr}[(G^+G^-)^n]$ as

$$\text{Tr}[(G^+G^-)^n] = \left(\frac{\ell}{4}\right)^{2n} \int_{[-\pi,\pi]^{2n}} \frac{d^{2n}k}{(2\pi)^{2n}} \int_{-1}^{1} d^{2n}\xi D(\{k\})F(\{k\})e^{i\ell\sum_{j=0}^{2n-1}\xi_{j+1}([k_{j+1}-k_j]_\pi)/2}. \tag{60}$$

Here $D(\{k\})$ is the same as in section 4, whereas $F(\{k_j\})$ is given by

$$F(\{k_j\}) = \left(\text{Tr}\prod_{j=0}^{n-1}\sigma_+^{(\ell k_{2j})}\sigma_-^{(\ell k_{2j+1})}\right)\left(\text{Tr}\prod_{j=0}^{2n-1}(a'\mathbb{1}_2 - b\sigma_{-i}^{(k_j)}e^{2it\varepsilon})\right). \tag{61}$$

To take the trace we use that

$$\sigma_+^{(\ell k_0)}\sigma_-^{(\ell k_1)}\cdots\sigma_+^{(\ell k_{2n-2})}\sigma_-^{(\ell k_{2n-1})} = e^{-i\ell\sum_{j=0}^{2n-1}k_j}\prod_{j=0}^{2n-1}(e^{ik_j\ell} + e^{ik_{j+1}\ell})\begin{pmatrix} e^{ik_{2n-1}\ell} & -i \\ ie^{i(k_0+k_{2n-1})\ell} & e^{ik_0\ell} \end{pmatrix}. \tag{62}$$

Notice that here we use periodic boundary conditions on the quasimomenta, meaning that $k_{2n} = k_0$. One can expand (62) to obtain

$$\text{Tr}\prod_{j=0}^{n-1}\sigma_+^{(\ell k_{2j})}\sigma_-^{(\ell k_{2j+1})} = 2 + \sum_{z=1}^{n}\sum_{j_1<\cdots<j_{2z}=0}^{2n-1}\left(e^{i\ell(k_{j_1}-k_{j_2}+k_{j_3}-\cdots-k_{j_{2z}})} + e^{-i\ell(k_{j_1}-k_{j_2}+k_{j_3}-\cdots-k_{j_{2z}})}\right). \tag{63}$$

Note the alternating pattern in the exponents in (63). The evaluation of the trace of the right term in (61) can be performed as in (44). The stationary phase approximation with respect to the variables $\zeta_j$ gives $[k_{j+1} - k_j]_\pi = 0$, i.e., the same as in (46). The trace in the right term in (61) is the same as in (45) after redefining $a \to a'$, and $n = 2p$. One obtains

$$2^{-2n}\text{Tr}\prod_{j=0}^{2n-1}\hat{t}''(k_j) = \sum_{j=0}^{\lfloor(2n-1)/2\rfloor}\binom{2n-1}{2j}(a')^{2n-2j}b^{2j}e^{2it\sum_{l=1}^{2j}\varepsilon(k_l)}$$

$$+ \sum_{j=0}^{p-1}\binom{2n-1}{2j+1}(a')^{2n-2j-2}b^{2j+2}e^{2it\sum_{l=0}^{2j+1}\varepsilon(k_l)}. \tag{64}$$

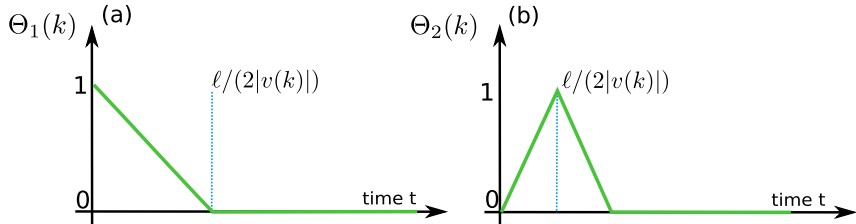

Figure 2: The functions $\Theta_1(k) = \max(0, 1 - 2|v(k)|t/\ell)$ (a) and $\Theta_2(k) = 2|v(k)|t/\ell + \max(2|v(k)|t/\ell, 2) - 2\max(2|v(k)|t/\ell, 1)$ (b) plotted versus time $t$. Here $v(k) = \varepsilon'(k)$ is the fermion group velocity (cf. (2)). The definitions are for the bipartition with two adjacent intervals of length $\ell$ (see Fig. 1).

It is now clear that when multiplying (63) and (64) the constant term in (63) can be treated as in section 4.1. In fact it gives the same result as (52) after replacing $a \to a'$ and $\ell \to 2\ell$. Importantly, there are additional terms that originate from the second term in (63). First, it is easy to check that the stationary phase approximation gives nonzero $\mu(\zeta_j)$ (cf. (41)) only when terms of (63) and (64) that contain the same quasimomenta $k_j$ are multiplied. Notice that the term with $j = 0$ in (64) does not contribute. It is also easy to check that within the stationary phase approximation all the contributions give the same $\mu(\{\zeta_j\}) \propto \Theta_2(k)$, with

$$\Theta_2(k) := 2|v(k)|t/\ell + \max(2|v(k)|t/\ell, 2) - 2\max(2|v(k)j|t/\ell, 1), \quad v(k) = \varepsilon'(k), \quad (65)$$

where we replaced $k_0 = k$. The absolute value $|v(k)|$ originates from the combination of the two terms in the sum in (63). The function $\Theta_2$ is pictorially defined in Fig. 2 (b). $\Theta_2(k)$ describes a linear growth up to $t = \ell/(2|v(k)|)$, which is followed by a linear decrease up to $t = \ell/(|v(k)|)$. At later times, $\Theta_2(k)$ is zero. Finally, the sum over $j$ in (64) gives

$$\text{Tr}[(G^+ G^-)^n] = \ell \int_{-\pi}^{\pi} \frac{dk}{2\pi} \left( 2(a')^{2n} + \left[ (a' - b)^{2n} + (a' + b)^{2p} - 2(a')^{2n} \right] \left( \Theta_1(k) + \frac{1}{2} \Theta_2(k) \right) \right). \tag{66}$$

The function $\Theta_1(k)$ is defined as

$$\Theta_1(k) := \max(0, 1 - 2|v(k)|t/\ell). \tag{67}$$

$\Theta_1(k)$ is plotted as a function of time in Fig. 2 (a). At $t = 0$, $\Theta_1(k) = 1$ and then it decreases linearly up to $t = \ell/(2|v(k)|)$. At later times $\Theta_1(k)$ is zero. Notice that $\Theta_1(k) + \Theta_2(k)/2 = \max(0, \ell - |v(k)|t/\ell)$. Thus, it is clear that Eq. (66) has the same structure as (51). Precisely, it is twice the result obtained from (52) after replacing $a \to 2a'$, $b \to 2b$, and $n \to 2n$. By using the formula

$$\text{Tr}(\mathbb{1}_{2\ell} + G^+ G^-)^{-1} = \sum_{p=0}^{\infty} \text{Tr}(-G^+ G^-)^p, \tag{68}$$

we obtain that (cf. (28))

$$\text{Tr}[(\mathbb{1}_{2\ell} + G^+ G^-)^{-1}] = \ell \int_{-\pi}^{\pi} \frac{dk}{2\pi} \left\{ \frac{2}{1 + (a')^2} \right.$$
$$\left. - \left[ \frac{(a' - b)^2}{1 + (a' - b)^2} + \frac{(a' + b)^2}{1 + (a' + b)^2} - \frac{2(a')^2}{1 + (a')^2} \right] \left( \Theta_1(k) + \frac{1}{2} \Theta_2(k) \right) \right\}. \tag{69}$$

To derive the term in the square brackets in (69) one has to remove the term with $p = 0$ in (68). This is clear from (66) because the term in the square brackets for $p = 0$ is zero.

## 5.2 Moments with defects insertions

Here we provide the formula describing the weak-dissipative hydrodynamic limit of the moments of the fermion correlation functions in the presence of defects insertions. The result is

$$
\mathrm{Tr}\left(\prod_{l=1}^{m}(G^{+}G^{-})^{q_l}G^{\alpha_l}\right)=\ell\int_{-\pi}^{\pi}\frac{dk}{2\pi}\bigg\{2(a')^{2s}+\big[(a'-b)^{2s}+(a'+b)^{2s}-2(a')^{2s}\big]\Theta_1
$$
$$
+\frac{1}{2}\big[(a'-b)^{m+\sum_l(2q_{2l-1}-d_{2l-1,2l})}(a'+b)^{\sum_l(2q_{2l}+d_{2l-1,2l})}
$$
$$
+(a'+b\leftrightarrow a'-b)-2(a')^{2s}\big]\Theta_2\bigg\}. \tag{70}
$$

Eq. (70) is obtained from the moments without defects $\mathrm{Tr}(G^{+}G^{-})^{n}$ by inserting the isolated operators $G^{\alpha_l}$. In (70) we defined $s=m/2+\sum_k q_k$ and $d_{i,j}$ as

$$
d_{i,j}:=\begin{cases}
1, & \text{for } (\alpha_i,\alpha_j)=(+,+),\\
1, & \text{for } (\alpha_i,\alpha_j)=(-,-),\\
0, & \text{for } (\alpha_i,\alpha_j)=(+,-),\\
2, & \text{for } (\alpha_i,\alpha_j)=(-,+).
\end{cases} \tag{71}
$$

The derivation of (70) is cumbersome, and we report the main steps in Appendix A. Eq. (70) gives access to several other moments constructed from the matrices $G^{\pm}$ and that are needed to obtain the fermionic negativity. For instance, by summing over $q_l$ in (70) we obtain

$$
\mathrm{Tr}\left(\prod_{l=1}^{m}(\mathbb{1}_{2\ell}+G^{+}G^{-})^{-1}G^{\alpha_l}\right)
$$
$$
=\ell\int_{-\pi}^{\pi}\frac{dk}{2\pi}\bigg\{2\left(\frac{a'}{1+(a')^2}\right)^m+\bigg[\left(\frac{a'-b}{1+(a'-b)^2}\right)^m+\left(\frac{a'+b}{1+(a'+b)^2}\right)^m-2\left(\frac{a'}{1+(a')^2}\right)^m\bigg]\Theta_1
$$
$$
+\frac{1}{2}\bigg[\frac{(a'+b)^{m-\sum_l d_{2l-1,2l}}}{[1+(a'+b)^2]^{m/2}}\frac{(a'-b)^{\sum_l d_{2l-1,2l}}}{[1+(a'-b)^2]^{m/2}}+(a'+b\leftrightarrow a'-b)-\frac{2(a')^m}{[1+(a')^2]^m}\bigg]\Theta_2\bigg\}. \tag{72}
$$

By summing over $\alpha_l$ in (72), we obtain

$$
\mathrm{Tr}\big[(G^{\mathrm{T}})^m\big]=\ell\int_{-\pi}^{\pi}\frac{dk}{2\pi}\bigg\{\left(\frac{1}{2}\pm\frac{a'}{1+(a')^2}\right)^m
$$
$$
+\frac{1}{2}\bigg[\left(\frac{1}{2}\pm\frac{a'-b}{1+(a'-b)^2}\right)^m+\left(\frac{1}{2}\pm\frac{a'+b}{1+(a'+b)^2}\right)^m-2\left(\frac{1}{2}\pm\frac{a'}{1+(a')^2}\right)^m\bigg]\Theta_1(k)
$$
$$
+\frac{1}{2}\bigg[\left(\frac{1}{2}\pm\frac{a'}{[1+(a'+b)^2]^{1/2}[1+(a'-b)^2]^{1/2}}\right)^m-\left(\frac{1}{2}\pm\frac{a'}{1+(a')^2}\right)^m\bigg]\Theta_2(k)\bigg\}. \tag{73}
$$

Here we removed the subscript $A$ in $G_A^{\mathrm{T}}$ to lighten the notation, although the correlation matrices are always restricted to subsystem $A$. In (73) one has to sum over the $\pm$ signs. As it is clear from (28), Eq. (73) is crucial to compute the logarithmic negativity.

## 5.3 Fermionic logarithmic negativity

We now have all the ingredients to discuss the quasiparticle picture for the fermionic logarithmic negativity. Before starting, we observe that (A.8) allows one to obtain the hydrodynamic

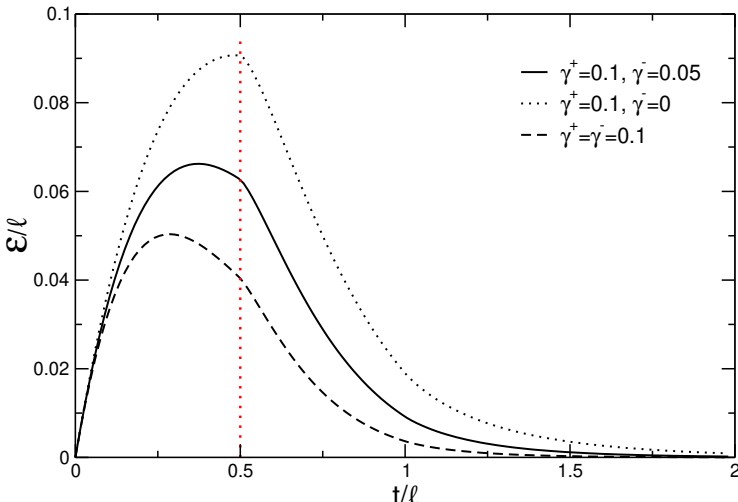

Figure 3: Dynamics of the fermionic negativity $\mathcal{E}$: Theoretical predictions in the weakly-dissipative hydrodynamic limit $t, \ell \to \infty$ $\gamma^{\pm} \to 0$ with $t/\ell$ and $\gamma^{\pm}\ell$ fixed. The results are for two adjacent intervals of equal length $\ell$. We plot $\mathcal{E}/\ell$ versus the rescaled time $t/\ell$ for several gain/loss rates $\gamma^{\pm}$. The negativity is smaller for balanced gain and losses, i.e., for $\gamma^{+} = \gamma^{-}$, and it increases upon increasing the imbalance between them. Notice the cusp-like singularity at $t/\ell = 1/2$ (marked by the vertical line), which reflects the presence of entangled quasiparticles.

behavior of $\text{Tr}[\mathcal{F}(C^{\text{T}})]$, for any $\mathcal{F}(z)$ that admits a Taylor expansion near $z = 0$. After expanding $\mathcal{F}(z)$, by applying (A.8) to all the terms, and resummming the series, we obtain that

$$
\begin{aligned}
\text{Tr}\big[\mathcal{F}(G^{\text{T}})\big] = \ell \int_{-\pi}^{\pi} \frac{dk}{2\pi} \bigg\{ &\mathcal{F}\bigg(\frac{1}{2} \pm \frac{1-a}{1+(1-a)^2}\bigg) \\
&+ \frac{1}{2}\bigg[\mathcal{F}\bigg(\frac{1}{2} \pm \frac{1-a-b}{1+(1-a-b)^2}\bigg) + \mathcal{F}\bigg(\frac{1}{2} \pm \frac{1-a+b}{1+(1-a+b)^2}\bigg) - 2\mathcal{F}\bigg(\frac{1}{2} \pm \frac{1-a}{1+(1-a)^2}\bigg)\bigg]\Theta_1(k) \\
&+ \frac{1}{2}\bigg[\mathcal{F}\bigg(\frac{1}{2} \pm \frac{1-a}{[1+(1-a+b)^2]^{1/2}[1+(1-a-b)^2]^{1/2}}\bigg) - \mathcal{F}\bigg(\frac{1}{2} \pm \frac{1-a}{1+(1-a)^2}\bigg)\bigg]\Theta_2(k)\bigg\},
\end{aligned}
\tag{74}
$$

where we replaced $a' = 1-a$, and where one has to sum over the $\pm$. To calculate the negativity (see first term in (28)) we have to use

$$
\mathcal{F}^{(1/2)}(z) := \ln(z^{1/2} + (1-z)^{1/2}).
\tag{75}
$$

We also need to calculate $\text{Tr}[\mathcal{F}^{(2)}(C_A)]$ where $C_A$ is the correlation matrix for $A_1 \cup A_2$ (cf. (5)) of length $2\ell$, with

$$
\mathcal{F}^{(2)}(z) := \frac{1}{2}\ln\big(z^2 + (1-z)^2\big).
\tag{76}
$$

The hydrodynamic prediction for the latter contribution is obtained from (53) as

$$
\begin{aligned}
\text{Tr}[\mathcal{F}^{(2)}(C_A)] = \ell \int_{-\pi}^{\pi} \frac{dk}{2\pi} \big[&2\mathcal{F}^{(2)}(a/2)\min(1, t/\ell|v(k)|) \\
&+ (\mathcal{F}^{(2)}((a+b)/2) + \mathcal{F}^{(2)}((a-b)/2))\max(0, 1 - t/\ell|v(k)|)\big].
\end{aligned}
\tag{77}
$$

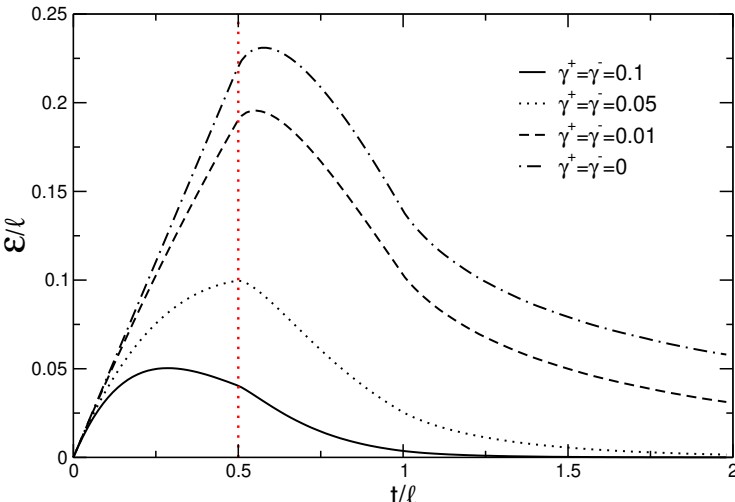

Figure 4: Dynamics of the fermionic negativity $\mathcal{E}$: Same as in Fig. 3 for balanced gain and losses, i.e., $\gamma^+ = \gamma^-$. We show the theoretical prediction for the negativity for $\gamma^+ = 0.1, 0.05, 0.01$. The dashed-dotted line is the prediction in the absence of dissipation, i.e., for $\gamma^+ = \gamma^- = 0$. In the absence of dissipation $\mathcal{E}$ exhibits a linear growth up to $t/\ell = 1/2$, followed by a "slow", i.e., power-law decay. Upon switching on the dissipation the initial linear growth is damped. Approximate linear growth can be observed only for $t \ll 1/\gamma^\pm$. Moreover, for finite $\gamma^\pm$, $\mathcal{E}$ vanishes exponentially at $t/\ell \to \infty$.

Let us consider the unitary limit $\gamma^\pm \to 0$. Thus, one obtains that $a \to 1$ and $b \to 1$. From (74), one has that

$$\mathrm{Tr}(\mathcal{F}^{(1/2)}(G_A^\mathrm{T})) = 2\ell \int_{-\pi}^{\pi} \frac{dk}{2\pi} \mathcal{F}(1/2)[1 - \Theta_1(k)], \tag{78}$$

where we used that $\mathcal{F}^{(1/2)}(0) = \mathcal{F}^{(1/2)}(1) = 0$. Importantly, as we also observed before, the term with $\Theta_2$ vanishes in the non-dissipative limit. On the other hand, we have

$$\mathrm{Tr}(\mathcal{F}^{(2)}(C_A)) = 2\ell \int_{-\pi}^{\pi} \frac{dk}{2\pi} \mathcal{F}^{(2)}(1/2) \min(1, |\nu(k)|t/\ell). \tag{79}$$

Putting together (78) (79) and (28), we obtain that

$$\mathcal{E} = \frac{\ell}{2} \int_{-\pi}^{\pi} \frac{dk}{2\pi} \ln(2)\Theta_2(k). \tag{80}$$

This is in agreement with the conjecture that $\mathcal{E} = I_{A_1:A_2}^{(1/2)}/2$ that was put forward in [23], where $I_{A_1:A_2}^{(1/2)}$ is the Rényi mutual information with Rényi index $1/2$ between the two intervals. This conjecture was also verified recently for the quench from the Néel state and the Majumdhar-Ghosh state in the $XX$ chain [84]. As we are going to discuss, the relation between negativity and Rényi mutual information is violated in the presence of dissipation.

Before discussing the final result for $\mathcal{E}$ we observe that

$$\mathcal{F}^{(1/2)}\left(\frac{1}{2} + \frac{1-a}{1+(1-a)^2}\right) + \mathcal{F}^{(1/2)}\left(\frac{1}{2} - \frac{1-a}{1+(1-a)^2}\right) + 2\mathcal{F}^{(2)}\left(\frac{a}{2}\right) = 0. \tag{81}$$

The left hand side of Eq. (81) would be the contribution to $\mathcal{E}$ for $t > \ell/|\nu(k)|$. Eq. (81) is consistent with the fact that the negativity is determined by the propagation of entangled

pairs of quasiparticles. Indeed, for $t > \ell/|v(k)|$ there are no entangled pairs shared between the two subsystems. Furthermore, within the quasiparticle picture the negativity should be proportional to $\Theta_2$, which is the number of entangled pairs that are shared between $A_1$ and $A_2$, as in the unitary case (cf. (80)). Indeed, a straightforward calculation shows that the negativity is given by

$$\mathcal{E} = \frac{\ell}{2} \int_{-\pi}^{\pi} \frac{dk}{2\pi} e(k) \Theta_2(k) , \qquad (82)$$

where

$$e := \ln\left( \frac{1}{2} \left( 1 - (1-a)^2 + b^2 + \sqrt{[1+(1-a)^2]^2 + 2[1-(1-a)^2]b^2 + b^4} \right) \right) , \qquad (83)$$

is the density of negativity. As anticipated, the function $\Theta_2(k)$ appears in (82). The structure of (82) is quite similar to the result for the logarithmic negativity after a global quantum quench in Conformal Field Theory [78]. Indeed, the same function $\Theta_2(k)$ appears. This gives the typical "rise and fall" dynamics of the negativity. Specifically, for two adjacent intervals of equal length $\ell$, $\mathcal{E}$ grows linearly up to time $t/(2v_{\max})$, with $v_{\max}$ the maximum velocity in the system. At asymptotically long times the negativity vanishes. In contrast with the behavior in Conformal Field Theory where the negativity decreases linearly up to $t = \ell/v_{\max}$, where it vanishes, Eq. (82) predicts a "slow" vanishing behavior. This is due to the fact that for lattice models the quasiparticles possess a nonlinear dispersion. The nonzero negativity at times $t > \ell/v_{\max}$ is due to slow quasiparticles.

Let us now discuss the negativity content of the quasiparticle pairs $e(k)$ in (82). First, we should observe that $e(k)$ depends on time because $a$ and $b$ are time-dependent (cf.(13) (15)). This is different from the CFT setup [78] and from the case of free-boson and free-fermion systems [23]. As a consequence of the factor $e^{-(\gamma^+ + \gamma^-)t}$ in the definition of $b$ (cf. (13)) in the presence of dissipation the linear growth at $t \leq \ell/(2v_{\max})$ is damped. Indeed, approximate linear behavior is visible only for $t \ll 1/(\gamma^+ + \gamma^-)$. Moreover, at long times $t \gg \ell$ the negativity decays to zero exponentially, in contrast with unitary dynamics, for which the decay is power law. Furthermore, in integrable free-fermions and free-boson systems, the negativity content of the quasiparticle pairs is the Rényi mutual information with Rényi index $1/2$. As it is clear from (82), this is not the case in the presence of dissipation.

Clearly, from (82) one recovers that in the non-dissipative case $\gamma^\pm = 0$, $e = \ln(2)$. Finally, it is interesting to focus on the balanced gain/loss dissipation. The condition $\gamma^+ = \gamma^-$ implies that $a = 1$, whereas $b = e^{-2\gamma^- t}$. This means that the term proportional to $\Theta_2(k)$ in (A.9) vanishes. Now, one has that $e(k) = \ln(1 + b^2)$. Interestingly, this implies that Eq. (83) is

$$e(k) = s_k^{(2),\text{YY}}(t) - s_k^{(2),\text{mix}}(t) , \qquad \text{for } \gamma^+ = \gamma^- , \qquad (84)$$

where $s_k^{(2),\text{YY}}$ and $s^{(2),\text{mix}}$ are the same as in (16). This means that

$$\mathcal{E} = \frac{1}{2} I_{A_1:A_2}^{(2)} , \qquad \text{for } \gamma^+ = \gamma^- , \qquad (85)$$

which makes apparent that in dissipative settings $\mathcal{E} \neq I^{(1/2)}/2$. We also verified that Eq. (85) does not remain valid for generic gain and loss processes.

One should also stress that the structure of (82) is quite revealing. Indeed, it is clear that Eq. (82) can be interpreted as the negativity of a few qubit systems. It should be possible to derive an effective few-qubits mixed density matrix describing the state of $A_1 \cup A_2$ that give the negativity in (82). Importantly, the effective density matrix is expected to be mixed because of the dissipative dynamics. The fact that the dynamics is dissipative is at the heart of the failure of the result of Ref. [23], which relies on the local dynamics being unitary. This idea

allows to understand the dynamics of the negativity free-fermions in the presence of localized losses [88]. Still, to derive the effective density matrix for the two intervals one would need at least a different quench in order to be able to guess how the elements of the matrix depend on the parameters of the system.

Our theoretical predictions for balanced gain and loss dissipation are reported in Fig. 4. In the figure we show results for vanishing dissipation rates $\gamma^+ = \gamma^- = 0.1, 0.05, 0.01, 0$. Clearly, as the dissipation is switched off, the unitary result is recovered. In particular, one recovers the linear increase up to $t = \mathcal{O}(\ell)$.

Finally, within the quasiparticle picture it is straightforward to generalize (82) to the case of two intervals at a distance $d$ (see Fig. 1). Indeed, as for the unitary case [23], it is natural to expect that the negativity content $e(k)$ of the entangled pairs remains the same as in (82), whereas only the function $\Theta_2(k)$ has to be modified. This is obtained by replacing $\Theta_2(k)$ in (82) with $\widetilde{\Theta}_2(k)$ defined as

$$\widetilde{\Theta}_2 := \max(2|v(k)|t/\ell, 2 + d/\ell) + \max(2|v(k)|t/\ell, d/\ell) - 2\max(2|v(k)|t/\ell, 1 + d/\ell). \quad (86)$$

Clearly, $\widetilde{\Theta}_2(k)$ appears in the quasiparticle picture for the mutual information between two intervals (see for instance [83]). This happens because both the mutual information and the negativity are proportional to the number of pairs shared between $A_1$ and $A_2$. $\widetilde{\Theta}_2(k)$ is zero for $t < d/(2|v(k)|)$. This reflects that at very short times there are entangled pairs that are shared between $A_1 \cup A_2$ and the rest, but there are no entangled pairs shared between $A_1$ and $A_2$ only. $\widetilde{\Theta}_2(k)$ grows linearly for $d/(2|v(k)|) \leq t < (d+\ell)/(2|v(k)|)$. At later times $\widetilde{\Theta}_2(k)$ decreases linearly. At any $t > (d+2\ell)/(2|v(k)|)$ it is identically zero.

We report analytical predictions for $\mathcal{E}$ for several dissipation rates $\gamma^\pm$ in Fig. 3 considering the case of adjacent intervals. For all the values of $\gamma^\pm$ the negativity exhibits the typical "rise and fall" behavior, with a growth at short times, followed by a vanishing behavior in the long time limit. The maximum at intermediate times is lower for the case with balanced gain/loss, i.e., for $\gamma^+ = \gamma^-$, and it progressively grows as the imbalance is increased. The vertical dashed line marks the point $t/\ell = 1/2$. All the curves exhibit a cusp-like singularity at this point, which reflects the presence of the entangled quasiparticles. Indeed, a similar cusp is present in the absence of dissipation.

Finally, let us stress again that Eq. (82) and Eq. (83) hold in the usual hydrodynamic limit with $\ell, d, t \to \infty$ with the ratios $t/\ell$ and $t/d$ fixed. Still, since $\mathcal{E}$ at fixed $\gamma^\pm$ vanishes exponentially as $e^{-(\gamma^+ + \gamma^-)t}$ for $t \to \infty$, it is convenient to take the weakly-dissipative hydrodynamic limit by sending $\gamma^\pm \to 0$ with fixed $\gamma^\pm \ell$.

## 6 Numerical benchmarks

Having derived the quasiparticle picture for the logarithmic negativity in the (weakly-dissipative) hydrodynamic limit, we now discuss some numerical checks. We first focus on several moments of the matrices $G^+ G^-$ in section 6.1. Finally, in section 6.2 we discuss numerical results for the negativity.

### 6.1 Moments of $G^+ G^-$

Let us discuss the moments

$$M_n := \text{Tr}\left[\prod_{p=1}^{n} G^{\alpha_p}\right], \quad \text{with} \quad \alpha_p = \pm. \quad (87)$$

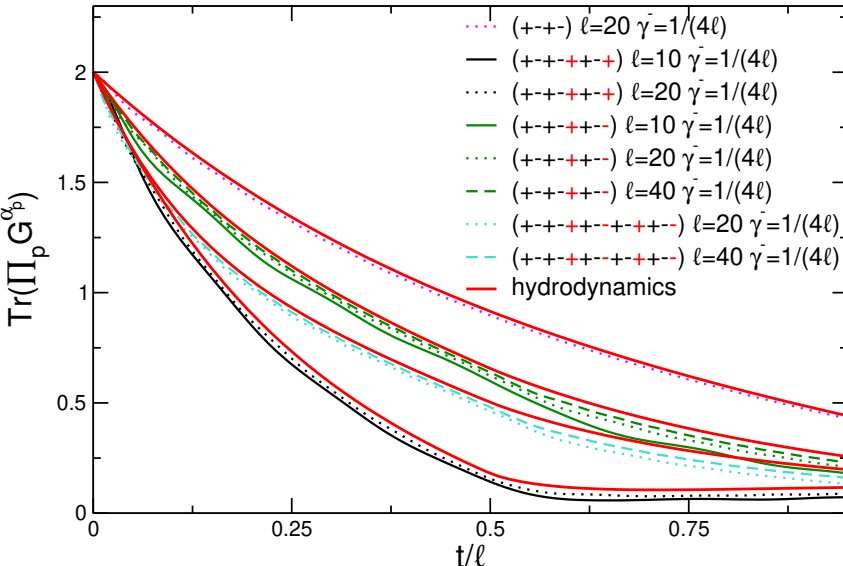

Figure 5: Dynamics of the moments $\mathrm{Tr}(\prod_p G^{\alpha_p})$ in the tight-binding chain with gain and loss dissipation. The results are for the quench from the fermionic Néel state and for two adjacent intervals of equal length $\ell$ (see Fig. 1). The string of $(\alpha_1, \alpha_2, \dots)$ that identifies the correlator is reported. Here we consider strings of operators $G^{\pm}$ with and without "defects", i.e., places where the same operator appears on consecutive sites (red symbols). We fix $\gamma^+ = 0$ and $\gamma^- = 1/(4\ell)$. The continuous red lines are the analytical results in the weakly-dissipative hydrodynamic limit $\ell, t \to \infty$ with $t/\ell$ and $\gamma^- \ell$ fixed. The theoretical results are obtained by using Eq. (66) and Eq. (70).

Here we focus on two adjacent intervals of length $\ell$. The correlator (87) is identified by the string $\{\alpha_1, \alpha_2, \dots, \alpha_n\}$. Our numerical results for $M_n$ are shown in Fig. 5. We only consider the situation with loss dissipation with rate $\gamma^-$. In Fig. 5 we consider both moments with defects insertions (see section 5.2), as well as without them. The operator insertions that create defects are denoted with red $\pm$ symbols. For each $M_n$ we consider several values of increasing $\ell$. To reach the weakly-dissipative hydrodynamic limit we fix $\gamma^- = 1/(4\ell)$. The analytic results in the scaling limit are reported as continuous red lines, and are obtained by using (70). For all the cases that we consider, at $t = 0$ we have $M_n = 2$ independently of $n$. Then, $M_n$ decrease, vanishing in the limit $t \to \infty$. It is important to stress that this is due to the fact that we have only loss dissipation. In the generic case with both gain and loss dissipation the behavior is different. Precisely, the moments start at $M_n = 2$, they exhibit a minimum at intermediate times, and saturate to a nonzero value at $t \to \infty$. As it is clear from Fig. 5, as we approach the weakly-dissipative hydrodynamic limit, deviations between the exact numerical data and the analytic predictions become progressively smaller. In Fig. 6 we consider the moments $M_n'$ defined as

$$M_n' := \mathrm{Tr}\Big[ \prod_{p=1}^{n} (\mathbb{1}_{2\ell} + G^+ G^-)^{-1} G^{\alpha_p} \Big]. \tag{88}$$

Similar to Fig. 5, we focus on $\gamma^+ = 0$. The red continuous lines are the results in the weakly-dissipative hydrodynamic limit. These are obtained by using (72). Already for moderately small $\gamma^-$ and large $t, \ell$ the data are in very good agreement with the analytical results. As a further check, in Fig. 7 we discuss the moments of $G^{\mathrm{T}}$ (cf. (28)). We report $\mathrm{Tr}[(G^{\mathrm{T}})^n]$ for $n = 3, 4$. We now consider gain dissipation only with $\gamma^+ = 1/(2\ell)$ and $\gamma^- = 0$. As it is clear from Fig. 7, already for $\ell = 10, 20$ the data are in excellent agreement with the analytical results in the weakly-dissipative hydrodynamic limit (continuous red lines) obtained from (73).

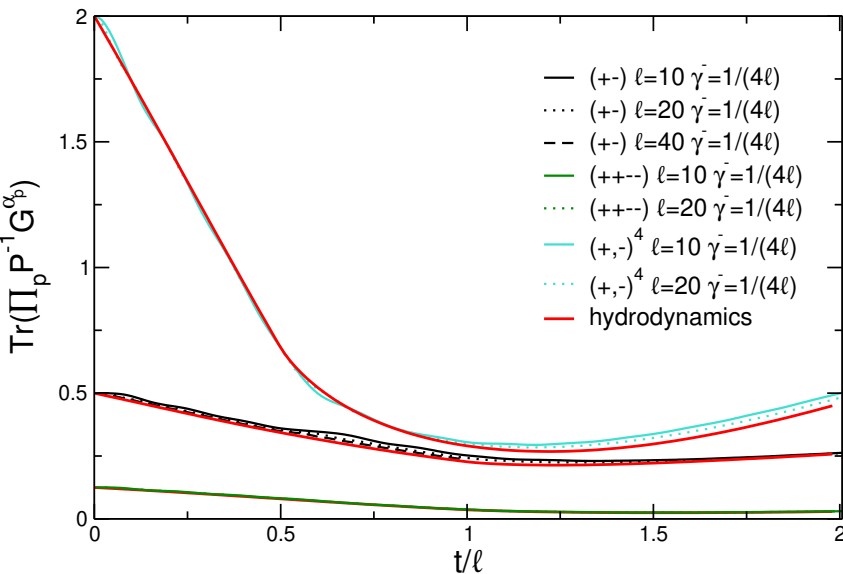

Figure 6: Dynamics of the moments $\text{Tr}\left(\prod_p P^{-1} G^{\alpha_p}\right)$ with $\alpha_p = \pm$ and $P = \mathbb{1}_{2\ell} + G^+ G^-$. Results are for the quench from the fermionic Néel state and for two adjacent intervals of equal length $\ell$. The gain and loss rates $\gamma^\pm$ are fixed as $\gamma^+ = 0$ and $\gamma^- = 1/(4\ell)$. In the legend we report as $(\alpha_1, \alpha_2, \dots)$ the configuration of the $\alpha_p$ that identify the different moments. Here $(+,-)^4$ denotes the sum over all the possible strings $(\alpha_1, \alpha_2, \alpha_3, \alpha_4)$. The continuous red lines are the results in the weakly-dissipative hydrodynamic limit $\ell, t \to \infty$ with $t/\ell$ and $\gamma^\pm \ell$ fixed. The analytical results are given by Eq. (72).

## 6.2 Logarithmic negativity

Let us finally discuss the dynamics of the fermionic negativity. We show numerical data for the rescaled fermionic negativity $\mathcal{E}/\ell$ plotted versus $t/\ell$ in Fig. 8. We now consider both gain and loss dissipation with rates $\gamma^+ = 1/(2\ell)$ and $\gamma^- = \gamma^+/2$. In Fig. 8 (a) we consider the situation with two adjacent intervals of equal length $\ell$, i.e., at distance $d = 0$ (see Fig. 1). In Fig. 8 (b) we focus on two disjoint intervals. Since we are interested in the hydrodynamic limit, we consider $d = \ell/2$. In the figure we show numerical data for $\ell = 10, 20, 40$. The data exhibit the typical "rise and fall" dynamics. For the two disjoint intervals (Fig. 8), $\mathcal{E} = 0$ for $t \leq d/(2v_{\max})$ with $v_{\max} = 1$. This is expected because for $t \leq d/(2v_{\max})$ there are no pairs of entangled quasiparticles that are shared between $A_1$ and $A_2$. Indeed, the first entangled pair contributing to the entanglement between the two intervals is created at a distance $d/2$ from them. The time $d/(2v_{\max})$ is the time at which the two quasiparticles forming the pair and traveling with $|v_{\max}| = 1$ reach $A_1$ and $A_2$, respectively. The quasiparticle prediction for the logarithmic negativity (cf. (79)) is reported in Fig. 8 as continuous red line. The agreement between (82) and the numerical data is remarkable for both adjacent and disjoint intervals.

## 7 Conclusions

We derived an exact formula for the dynamics of the fermionic logarithmic negativity after the quench from the fermionic Néel state in the tight-binding chain with both gain and loss dissipation. Our main result is formula (82). As a byproduct we provided analytical results for several fermionic correlators. Formula (82) shows that the negativity admits a quasiparticle

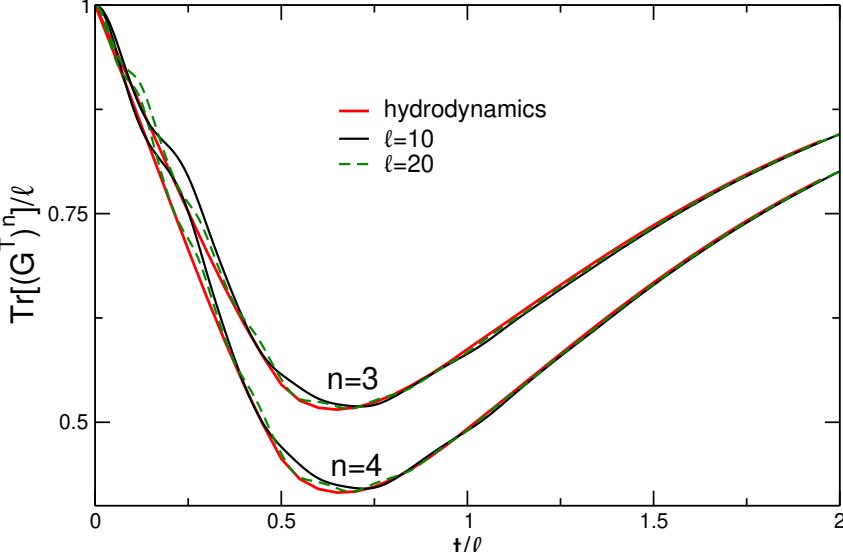

Figure 7: Dynamics of the moments $\text{Tr}[(G^T)^n]$ in the tight-binding chain with gain and loss dissipation. The results are for two adjacent intervals of equal length $\ell$ (see Fig. 1). Dissipation rates $\gamma^\pm$ are chosen as $\gamma^+ = 1/(2\ell)$ and $\gamma^- = 0$. We show the rescaled moments $\text{Tr}(G^T)^n/\ell$ versus $t/\ell$. The results are for $n = 2, 3$ and $\ell = 10, 20$. The continuous red line is the the weakly-dissipative hydrodynamic limit $t, \ell \to \infty$, with $t/\ell$ and $\gamma^\pm\ell$ fixed. The analytical results are given by Eq. (73).

picture interpretation. Similar to the mutual information, the negativity is proportional to the number of entangled pairs that are shared between two intervals. This is reflected in its typical "rise and fall" dynamics. Still, the negativity content of the quasiparticles originates from an intricate interplay between unitary and dissipative contributions. In particular, the negativity is not easily related to standard thermodynamic quantities. This is in constrast with what happens for the mutual information, which can be related to the thermodynamic entropy of the system [8, 9]. Moreover, our result shows explicitly that in the presence of dissipation the logarithmic negativity is not half of the Rényi mutual information with Rényi index 1/2, in contrast with the unitary case [23].

Let us now mention some interesting future directions. First, it would be important to extend the quasiparticle picture for the negativity to other quenches and other free-fermion systems. Indeed, it is likely that a formula for generic quenches and quadratic dissipation can be obtained. A good starting point would be to consider the quench from the Majumdar-Ghosh state in the tight-binding chain [89]. Another interesting direction would be to consider out-of-equilibrium dynamics in bosonic systems [8]. Moreover, it would be interesting to study the negativity in the presence of localized dissipations. Indeed, it has been shown in Ref. [90] that the dynamics of the von Neumann and the Rényi entropies in the presence of localized fermion losses are determined by the effective transmission and reflection coefficients of the lossy site. It would be interesting to understand how to generalize this result to the negativity. Finally, an important open problem is to understand the behavior of the logarithmic negativity in dissipative interacting integrable systems.

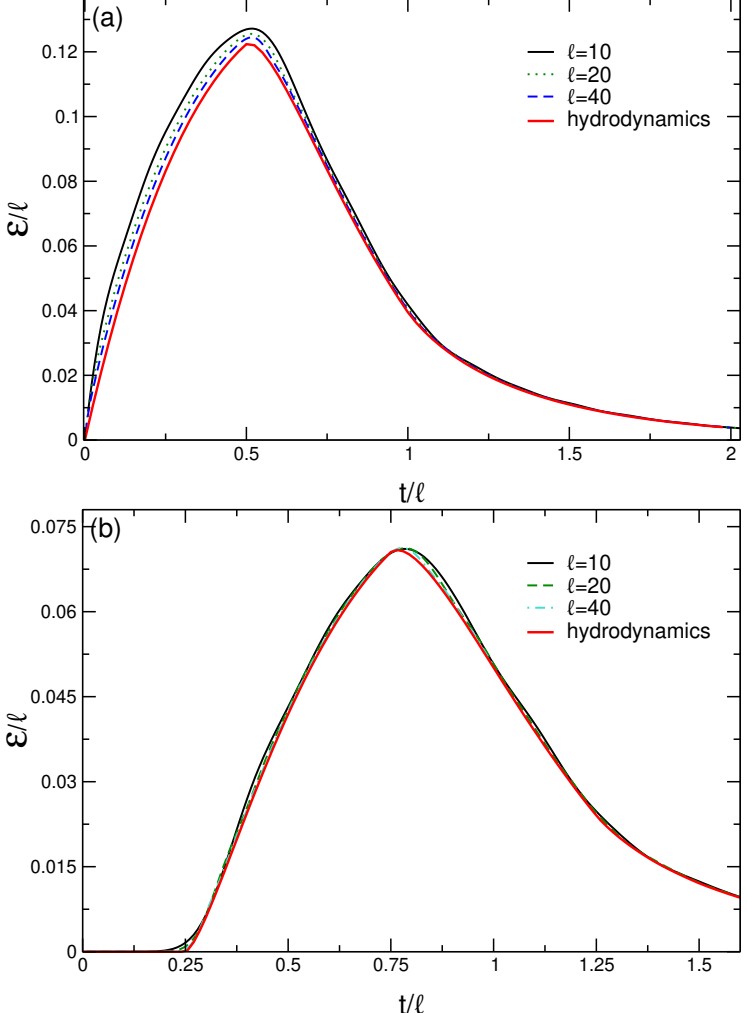

Figure 8: Dynamics of the fermionic negativity $\mathcal{E}$ after the quench from the fermionic Néel state in the tight-binding chain with gain/loss dissipation. Results are for the negativity between two intervals of equal length $\ell$. The data are for $\gamma^+ = 1/(2\ell)$ and $\gamma^- = \gamma^+/2$. The figure shows the scaling plot of $\mathcal{E}/\ell$ versus $t/\ell$. The continuous red line is the result in the weakly-dissipative hydrodynamic limit. In (a) we show results for two adjacent intervals, whereas in (b) we discuss the case of two disjoint intervals at $d = \ell/2$. The analytical results are given by (82), where for disjoint intervals we replaced $\Theta_2 \to \widetilde{\Theta}_2$ (cf. (86)).

# Acknowledgements

**Funding information** F.C. acknowledges support from the "Wissenschaftler- Rückkehrprogramm GSO/CZS" of the Carl-Zeiss-Stiftung and the German Scholars Organization e.V., as well as through the Deutsche Forschungsgemeinsschaft (DFG, German Research Foundation) under Project No. 435696605, as well as through the Research Unit FOR 5413/1, Grant No. 465199066. F.C. is indebted to the Baden-Württemberg Stiftung for the financial support by the Eliteprogramme for Postdocs.

# A   Moments of $G^+ G^-$ with insertions of defects

In order to calculate the moments of $G^{\mathrm{T}}$ (cf. (28)) one has to deal with terms of the form

$$\mathrm{Tr}\left(\prod_{l=1}^{m}(G^+ G^-)^{q_l} G^{\alpha_l}\right), \quad \text{with} \quad \alpha_l = \pm. \tag{A.1}$$

Here $q_l$ is a positive integer. The term $(G^+ G^-)^{q_l}$ is obtained by using (68) to expand $(\mathbb{1}_{2\ell} + G^+ G^-)^{-1}$. The term under the trace in (A.1) is obtained by breaking the alternating pattern in $(G^+ G^-)^{\sum_{l=1}^{m} q_l}$ via the insertion of $m$ "misplaced" matrices (defects) $G^{\alpha_l}$ at positions $2q_l + l$. Now, following the approach in section 5.1, one obtains an expression similar to (61). In particular, the presence of the defects in (A.1) does not affect the second trace in (61), which depends only on the details of the quench and of the Hamiltonian. The term inside the first trace in (61) has to be modified, although in a simple manner. Specifically, some of the terms $e^{ik_j \ell} + e^{ik_{j+1}\ell}$ in the product in (62) get a relative minus sign.

Before considering the generic situation with arbitrary $m$, it is useful to focus on $m = 2$. The case with $m = 1$ can be neglected because we numerically observe that Eq. (A.1) vanishes in the hydrodynamic limit for any odd $m$. For now, let us consider $m = 2$ and $\alpha_1 = +$ and $\alpha_2 = -$. One can use (62) to obtain

$$\mathrm{Tr}[(G^+ G^-)^{q_1} G^{\alpha_1} (G^+ G^-)^{q_2} G^{\alpha_2}] = e^{-i\ell \sum_{j=1}^{2q_1+2q_2+2} k_{j-1}} \prod_{j=1}^{2q_1+2q_2+2} \left(e^{ik_{j-1}\ell} + s_j e^{ik_j \ell}\right), \tag{A.2}$$

where $s_j = 1$ except for the sites near the positions of the defects. Precisely, $s_j = -1$ if a "misplaced" $G^-$ is inserted at $j+1$, and $s_j = -1$ if $G^+$ is inserted at $j$. Clearly, Eq. (A.2) can be generalized to account for generic $\alpha_l$. A straightforward although tedious calculation allows one to obtain that

$$\mathrm{Tr}[(G^+ G^-)^{q_1} G^+ (G^+ G^-)^{q_2} G^-] = \ell \int_{-\pi}^{\pi} \frac{dk}{2\pi} \Big\{ 2(a')^{2s} + \big[(a'-b)^{2s} + (a'+b)^{2s} - 2(a')^{2s}\big] \Theta_1(k)$$

$$+ \frac{1}{2}\big[(a'-b)^{2(q_1+1)}(a'+b)^{2q_2} + (a'+b)^{2(q_1+1)}(a'-b)^{2q_2} - 2(a')^{2s}\big]\Theta_2(k)\Big\}, \tag{A.3}$$

where we defined $s := q_1 + q_2 + 1$. Interestingly, the first two terms do not contain information about the defects. In fact they coincide with the first two terms in (66) after changing $n \to q_1 + q_2 + 1$. They depend only on the total number of operators $G^{\pm}$ present in (A.1). On the other hand, the term multiplying $\Theta_2(k)$ (third term in (A.3)) depends on the defects. This term is obtained from the second one by replacing $(a'-b)^{2q_1+2q_2+1} \to (a'-b)^{2q_1+2}(a'+b)^{2q_2}$ and $(a'+b)^{2q_1+2q_2+2} \to (a'+b)^{2q_1+2}(a'-b)^{2q_2}$. The change in the relative sign between $a'$ and $b$ reflects the presence of the defects in (A.1). A similar structue is present for generic $\alpha_1, \alpha_2$. We verified that

$$\mathrm{Tr}\left(\prod_{l=1}^{2}(G^+ G^-)^{q_l} G^{\alpha_l}\right) = \ell \int_{-\pi}^{\pi} \frac{dk}{2\pi} \Big\{ 2(a')^{2s} + \big[(a'-b)^{2s} + (a'+b)^{2s} - 2(a')^{2s}\big] \Theta_1(k)$$

$$+ \frac{1}{2}\big[(a'-b)^{2(q_1+1)-d_{1,2}}(a'+b)^{2q_2+d_{1,2}} + (a'+b \leftrightarrow a'-b) - 2(a')^{2s}\big]\Theta_2(k)\Big\}. \tag{A.4}$$

Here we defined $d_{i,j}$ as

$$d_{i,j} := \begin{cases} 1, & \text{for } (\alpha_i, \alpha_j) = (+,+), \\ 1, & \text{for } (\alpha_i, \alpha_j) = (-,-), \\ 0, & \text{for } (\alpha_i, \alpha_j) = (+,-), \\ 2, & \text{for } (\alpha_i, \alpha_j) = (-,+). \end{cases} \tag{A.5}$$

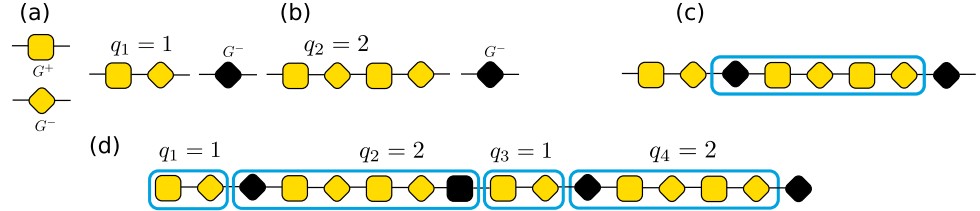

Figure 9: Pictorial illustration of the operators $G^\pm$ (a). In (b) we show the pictorial definition of $(G^+G^-)^{q_1}G^-(G^+G^-)^{q_2}G^-$ with $q_1 = 1$ and $q_2 = 2$. In (c) we show the effect of two defects due to the insertion of two misplaced operators (black symbols). The number of operators between the two defects is $2q_2 + d_{12} = 5$. (d) The case with four defects with $q_1 = q_3 = 1$ and $q_2 = q_4 = 2$. The contribution of the configuration to the last term in (A.6) is $(a'+b)^{m+2q_1+2q_3-d_{12}-d_{34}}(a'-b)^{2q_2+2q_4+d_{12}+d_{34}}$ plus the term with $a' - b$ and $a' + b$ exchanged.

Let us now discuss the case with generic $m$. We verified that formula (A.4) can be generalized to arbitrary number of defects as

$$
\begin{aligned}
\operatorname{Tr}\left(\prod_{l=1}^{m}(G^+G^-)^{q_l}G^{\alpha_l}\right) &= \ell \int_{-\pi}^{\pi}\frac{dk}{2\pi}\Big\{2(a')^{2s} + \Big[(a'-b)^{2s} + (a'+b)^{2s} - 2(a')^{2s}\Big]\Theta_1 \\
&+ \frac{1}{2}\Big[(a'-b)^{m+\sum_l(2q_{2l-1}-d_{2l-1,2l})}(a'+b)^{\sum_l(2q_{2l}+d_{2l-1,2l})} + (a'+b \leftrightarrow a'-b) - 2(a')^{2s}\Big]\Theta_2\Big\},
\end{aligned}
\tag{A.6}
$$

with $d_{i,j}$ defined in (71) and $s := m/2 + \sum_k q_k$. Again, as for the case with $m = 2$ (cf. (A.3)) the term multiplying $\Theta_1(k)$ does not depend on the defects insertions. Oppositely, the term multiplying $\Theta_2$ contains information about the defects. The structure of this term is illustrated in Fig. 9. In Fig. 9 (a) we denote with a square and a diamond the two operators $G^+$ and $G^-$, respectively. In (b) we show the multiplication of the string of operators with $m = 2$, $q_1 = 1$, $q_2 = 2$ and $\alpha_1 = -, \alpha_2 = -$. The result is shown in Fig. 9 (c). Defects are now present at places where the same operator is on consecutive sites. The box encloses the operators in between two defects. Notice that the number of operators in the box is $2q_2 + d_{12} = 5$ (cf. Eq. (71)). Similarly, one can recover the other cases of Eq. (71) by considering other values of $\alpha_1, \alpha_2$. A more complicated contraction with $m = 4$ is shown in Fig. 9 (d). Now we have $\{\alpha_1, \alpha_2, \alpha_3, \alpha_4\} = \{-, +, -, -\}$ and $\{q_1, q_2, q_3, q_4\} = \{1, 2, 1, 2\}$. In the last term in (A.6), this corresponds to $(a'+b)^{4+2q_1+2q_3-d_{1,2}-d_{3,4}}(a-b)^{2q_2+2q_4+d_{1,2}+d_{2,4}}$ plus the term with the relative sign between $a'$ and $b$ exchanged. After summing over $q_l$, we obtain

$$
\begin{aligned}
\operatorname{Tr}\left(\prod_{l=1}^{m}(\mathbb{1}_{2\ell} + G^+G^-)^{-1}G^{\alpha_l}\right) &= \ell \int_{-\pi}^{\pi}\frac{dk}{2\pi}\Bigg\{2\left(\frac{a'}{1+(a')^2}\right)^m + \Bigg[\left(\frac{a'-b}{1+(a'-b)^2}\right)^m + \left(\frac{a'+b}{1+(a'+b)^2}\right)^m - 2\left(\frac{a'}{1+(a')^2}\right)^m\Bigg]\Theta_1 \\
&+ \frac{1}{2}\Bigg[\frac{(a'+b)^{m-\sum_l d_{2l-1,2l}}}{[1+(a'+b)^2]^{m/2}}\frac{(a'-b)^{\sum_l d_{2l-1,2l}}}{[1+(a'-b)^2]^{m/2}} + (a'+b \leftrightarrow a'-b) - \frac{2(a')^m}{[1+(a')^2]^m}\Bigg]\Theta_2\Bigg\}.
\end{aligned}
\tag{A.7}
$$

After summing over $\alpha_l = \pm$, we obtain

$$\mathrm{Tr}\left(\sum_{\{\alpha_l=\pm\}}\prod_{l=1}^{m}(\mathbb{1}_{2\ell}+G^+G^-)^{-1}G^{\alpha_l}\right)=2^{m+1}\ell\int_{-\pi}^{\pi}\frac{dk}{2\pi}\left\{\left(\frac{a'}{1+(a')^2}\right)^m\right.$$

$$+\frac{1}{2}\left[\left(\frac{a'-b}{1+(a'-b)^2}\right)^m+\left(\frac{a'+b}{1+(a'+b)^2}\right)^m-2\left(\frac{a'}{1+(a')^2}\right)^m\right]\Theta_1(k)$$

$$+\frac{1}{2}\left[\frac{(a')^m}{[1+(a'+b)^2]^{m/2}[1+(a'-b)^2]^{m/2}}-\frac{(a')^m}{[1+(a')^2]^m}\right]\Theta_2(k)\right\}. \quad \text{(A.8)}$$

Curiously, the term multiplying $\Theta_2(k)$ vanishes in the non-dissipative limit $a' \to 0$, despite the fact that it shows the "rise and fall" dynamics expected for the negativity (see Fig. 2 (b)). From (A.8) we now obtain the moments of $G^{\mathrm{T}}$ (cf. (28)) as

$$\mathrm{Tr}\left[(G^{\mathrm{T}})^m\right]=\ell\int_{-\pi}^{\pi}\frac{dk}{2\pi}\left\{\left(\frac{1}{2}\pm\frac{a'}{1+(a')^2}\right)^m\right.$$

$$+\frac{1}{2}\left[\left(\frac{1}{2}\pm\frac{a'-b}{1+(a'-b)^2}\right)^m+\left(\frac{1}{2}\pm\frac{a'+b}{1+(a'+b)^2}\right)^m-2\left(\frac{1}{2}\pm\frac{a'}{1+(a')^2}\right)^m\right]\Theta_1(k)$$

$$+\frac{1}{2}\left[\left(\frac{1}{2}\pm\frac{a'}{[1+(a'+b)^2]^{1/2}[1+(a'-b)^2]^{1/2}}\right)^m-\left(\frac{1}{2}\pm\frac{a'}{1+(a')^2}\right)^m\right]\Theta_2(k)\right\}, \quad \text{(A.9)}$$

where one has to sum over the ±. Again, as for (A.8), the term multiplying $\Theta_2(k)$ vanishes in the nondissipative limit.

# B Logarithmic negativity for particle-number-conserving free-fermion systems

In this appendix we report the derivation of formula (29) for the fermionic logarithmic negativity [21] for free-fermion systems with fixed fermion number. Specifically, fermion number conservation implies

$$\langle c_j c_l\rangle=\langle c_j^\dagger c_l^\dagger\rangle=0,\quad \forall\, j,l\,. \quad \text{(B.1)}$$

As a consequence of Eq. (B.1), the negativity can be expressed in terms of the correlation matrix $C_{jl}$ defined as

$$C_{jl}:=\langle c_j^\dagger c_l\rangle\,. \quad \text{(B.2)}$$

In order to show that, we start from the more general definition of the negativity in terms of Majorana correlation functions, which holds true also for generic, i.e., non particle-conserving fermion systems [21]. Let us define the Majorana operators $a_j$ as

$$c_j:=\frac{1}{2}\left(a_{2j-1}-ia_{2j}\right),\quad c_j^\dagger:=\frac{1}{2}(a_{2j-1}+ia_{2j})\,. \quad \text{(B.3)}$$

Here $a_j$ satisfy the standard anticommutation relations

$$\{a_j,a_l\}=2\delta_{jl}\,. \quad \text{(B.4)}$$

In the following we are going to assume that (B.1) holds in the initial state and at any time. From the definitions (B.2) and (B.3) we obtain that

$$C_{jl}=\frac{1}{4}\left(a_{2j-1}a_{2l-1}-ia_{2j-1}a_{2l}+ia_{2j}a_{2l-1}+a_{2j}a_{2l}\right)\,. \quad \text{(B.5)}$$

After using (B.1), Eq. (B.5) becomes

$$C_{jl} = \frac{1}{2}\left(a_{2j}a_{2l} + ia_{2j}a_{2l-1}\right). \tag{B.6}$$

This implies that

$$2\text{Re}(C_{jl}) = \delta_{jl} + ia_{2j}a_{2l-1}, \qquad 2i\text{Im}(C_{jl}) = a_{2j}a_{2l} - \delta_{jl}. \tag{B.7}$$

Let us now define the Majorana correlation matrix $\Gamma_{jl}$ as

$$\Gamma_{jl} := \frac{1}{2}\langle[a_j, a_l]\rangle = \langle a_j a_l\rangle - \delta_{jl}. \tag{B.8}$$

From (B.7) we obtain that

$$\Gamma_{2j,2l} = \Gamma_{2j-1,2l-1} = 2i\text{Im}(C_{jl}), \tag{B.9}$$

$$\Gamma_{2j-1,2l} = -\Gamma_{2j,2l-1} = -i\left(2\text{Re}(C_{jl}) - \delta_{jl}\right). \tag{B.10}$$

The correlation matrix $G_{jl}$ (cf. (25)) is obtained as

$$G_{jl} := 2C_{jl} - \delta_{jl} = \Gamma_{2j,2l} + i\Gamma_{2j,2l-1}. \tag{B.11}$$

Let us now consider the $2\ell \times 2\ell$ correlation matrix $G$ restricted to subsysytem $A$, i.e., with $j,l \in A$ (see Fig. 1). We also define the $4\ell \times 4\ell$ restricted Majorana correlation matrix as

$$\Gamma = \begin{pmatrix} \Gamma_{2j,2l} & \Gamma_{2j-1,2l} \\ -\Gamma_{2j-1,2l} & \Gamma_{2j,2l} \end{pmatrix}, \tag{B.12}$$

where we used (B.9) and (B.10). The eigenvalues and eigenvectors of $\Gamma$ are simply related to those of $G_{jl}$. To show that, let us consider a generic eigenvalue $\lambda$ of $G_{jl}$ with eigenvector $v_j$. From (B.11) one has that

$$(\Gamma_{2j,2l} + i\Gamma_{2j,2l-1})v_l = \lambda v_j, \tag{B.13}$$

where the sum over repeated indices is assumed. Now, one can check that

$$\Gamma v_+ = \lambda v_+, \quad \Gamma v_- = -\bar{\lambda} v_-, \quad \text{with} \quad v_+ = \begin{pmatrix} v_j \\ -iv_j \end{pmatrix}, \quad v_- = \begin{pmatrix} \bar{v}_j \\ i\bar{v}_j \end{pmatrix}, \tag{B.14}$$

where $v_j$ are the components of the eigenvectors of $G_{jl}$ (cf. (B.13)) and the bar in $\bar{v}_j$ denotes the complex conjugate. To verify (B.14) one has to use that $\bar{v}_j$ satify

$$(\Gamma_{2j,2l} - i\Gamma_{2j,2l-1})\bar{v}_l = \bar{\lambda}\bar{v}_j, \tag{B.15}$$

which is obtained by taking the complex conjugate of (B.13) and by using that $\bar{\Gamma}_{2j,2l} = -\Gamma_{2j,2l}$ and $\bar{\Gamma}_{2j,2l-1} = -\Gamma_{2j,2l-1}$. Furthermore, here we notice that $\lambda$ is real, because $G_{jl}$ is an hermitian matrix. This means that given the eigenvalues $\lambda_k$ of $G_{jl}$, the eigenvalues of $\Gamma$ are organized in pairs as $(\lambda_k, -\lambda_k)$.

A similar result holds for the matrices $\Gamma^{\pm}$ (cf. (27)). Let us first define $\Gamma^{\pm}$ as

$$\Gamma^{\pm} = \begin{pmatrix} \Gamma^{11} & \pm i\Gamma^{12} \\ \pm i\Gamma^{21} & -\Gamma^{22} \end{pmatrix}. \tag{B.16}$$

Here $\Gamma^{pq}$ ($p,q = 1,2$) are defined in (B.12) with the constraint that $j \in A_p$ and $l \in A_q$. The eigenvalues and eigenvectors of $\Gamma^{\pm}$ are simply related to those of $G^{\pm}$ (cf. (27)) defined as

$$G^{\pm} = \begin{pmatrix} G^{11} & \pm iG^{12} \\ \pm iG^{21} & -G^{22} \end{pmatrix}. \tag{B.17}$$

First, we observe that $G^{\pm}$ and $\Gamma^{\pm}$ are not hermitian, implying that they have different left and right eigenvectors. Let us consider the right eigenvector $w_j^{\pm}$ ($j = 1, \ldots, 2\ell$) of $G^{\pm}$ with eigenvalue $\mu^{\pm}$. One can show by direct computation that the right eigenvector $V_{\mu^{\pm}}$ of $\Gamma^{\pm}$ with eigenvalue $\mu^{\pm}$ is obtained as

$$V_{\mu^{\pm}} = \left\{ w_1^{\pm}, -iw_1^{\pm}, w_2^{\pm}, -iw_2^{\pm}, \ldots, w_{2\ell}^{\pm}, -iw_{2\ell}^{\pm} \right\}. \tag{B.18}$$

The eigenvector $V_{-\bar{\mu}^{\pm}}$ associated with the other eigenvalue $-\bar{\mu}^{\pm}$ of $\Gamma^{\pm}$ is given as

$$V_{-\bar{\mu}^{\pm}} = \left\{ \bar{w}_1^{\pm}, -i\bar{w}_1^{\pm}, \bar{w}_2^{\pm}, -i\bar{w}_2^{\pm}, \ldots, \bar{w}_{\ell}^{\pm}, -i\bar{w}_{\ell}^{\pm}, -\bar{w}_{\ell+1}^{\pm}, i\bar{w}_{\ell+1}^{\pm}, \ldots, -\bar{w}_{2\ell}^{\pm}, i\bar{w}_{2\ell}^{\pm} \right\}. \tag{B.19}$$

Finally, in a similar way one can obtain the spectrum of $\Gamma^T$ defined as

$$\Gamma^T = \frac{1}{2} \left[ \mathbb{1}_{4\ell} - (\mathbb{1}_{4\ell} + \Gamma^+ \Gamma^-)^{-1}(\Gamma^+ + \Gamma^-) \right], \tag{B.20}$$

from that of $G^T$ defined as (cf. (28))

$$G^T = \frac{1}{2} \left[ \mathbb{1}_{2\ell} - (\mathbb{1}_{2\ell} + G^+ G^-)^{-1}(G^+ + G^-) \right]. \tag{B.21}$$

First, both $\Gamma^T$ and $G^T$ are hermitian matrices, and hence have real eigenvalues. Now, let us consider the eigenvector $Z = \{z_1, \ldots, z_{2\ell}\}$ of $\mathbb{1}_{2\ell}/2 - G^T$ with eigenvalue $\zeta$. One can show that $\mathbb{1}_{4\ell}/2 - \Gamma^T$ has eigenvalues $\pm\zeta$. The eigenvectors are obtained from (B.18) and (B.19) after replacing $w_j \to z_j$. Thus, the eigenvalues $\nu_i$ of $\Gamma^T$ are given as $\nu_i = (\xi_i, 1 - \xi_i)$, with $\xi_i$ the eigenvalues of $G^T$. Finally, for a generic free-fermion system the negativity is obtained as [50]

$$\mathcal{E} = \frac{1}{2} \sum_{i=1}^{4\ell} \ln(\nu_i^{1/2} + (1 - \nu_i)^{1/2}) - \frac{1}{2} S_A^{(2)}, \tag{B.22}$$

where $\nu_i$ are the eigenvalues of $\Gamma^T$, and $S_A^{(2)}$ is the second Rényi entropy of $A = A_1 \cup A_2$ (see Fig. 1). Given the relationship $\nu_i = (\xi_i, 1 - \xi_i)$ between $\nu_i$ and the eigenvalues $\xi_i$ of $G^T$, it is clear that (B.22) is the same as (29).

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
