# Peer review of "Logarithmic negativity in out-of-equilibrium open free-fermion chains: An exactly solvable case"

_SciPost Physics, doi:SciPost Phys. 15, 124 (2023)_

## Round 1 · Referee Report · Anonymous (Referee 1) · 2022-7-9

Report

The authors study the dynamics of logarithmic negativity in an open free-fermion chain. They consider a tight-binding model with the addition of on-site dissipation. After introducing the model, and the so-called "fermionic negativity" (a quantity closely related to negativity that is easier to calculate in free-fermionic systems), they derive expressions describing the dynamics of Renyi entropies and negativity, valid in the special scaling limit of subsystem sizes and time going to infinity with fixed ratios, which they refer to as the "hydrodynamic limit", while the dissipative rates need to scale as the inverse of the subsystem size (referred to as "weakly-dissipative" limit). Interestingly, they find that the usual relation between the Renyi-1/2 mutual information and negativity ceases to hold. Surprisingly, in certain limits they recover an analogous relation between negativity and Renyi-2 mutual information. They also provide numerical data that supports the analytical predictions.

I believe that the main results of the paper are interesting and deserve publication, but not in the current form. Derivations are hard to follow up to the point of the paper being almost unreadable. I am aware that this is a calculation-heavy paper, and is therefore technical in nature. However, the authors should make a stronger effort to make the paper more accessible. I propose the authors try to clean the calculations up, add explanations where relevant, and maybe move some of the more technical parts to an appendix. Moreover, the sections preceding the derivation are missing some more intuitive explanation of the physics discussed there. It feels like a recount of previous results by the authors without enough context to make the paper self contained. I do not expect the authors to rederive all the previous results leading to the calculation in Sec. 5, but it would be nice to get an idea of where most of the statements come from without having to be thouroughly familiar with an extensive list of literature.

Furthermore, I find that the writing is at certain points sloppy with some statements that are misleading or even plainly false. Below I'm attaching a list of examples that I found.

1st paragraph:
"While for bipartite quantum systems in a pure state several quantum information motivated measures can be used to identify entanglement [1-4], this is not the case if the state of the full system is mixed." <- This seems to be incorrect: the authors in the next paragraph introduce a measure for mixed-state entanglement, which is also motivated by quantum information.

Page 2, 2nd paragraph:
"For interacting integrable systems no results are available." <- This is again not true. Ref. [80] suggests that in the early-time regime and for contiguous tripartition this relation between Renyi-1/2 mutual information and logarithmic negativity holds exactly (i.e. not only in the scaling limit). This result holds for any local quantum circuit, and therefore also for interacting integrable systems. See also Ref. [75] for a similar claim in the case of CFTs.

"Despite this scenario, it is possible to obtain analytically the dynamics of the logarithmic negativity in the rule 54 chain, and it is in agreement with Ref. [23]." <- Again, this is misleading. As far as I'm aware there is no explicit calculation for Rule 54. The only result that could come close to that is the general statement of Ref. [80] that holds for *all* local unitary circuits (see above), but this is only in agreement with Ref. [23] in the early-time regime.

Sec. 2: I am assuming that L is even. It could be useful to point it out explicitly.

Eq. (5): It could be useful to define what kind of expectation value it is, i.e.
C_{jl}:=<c_j c_l^{\dagger}>=<\Psi_0|c_j^{\dagger}c_l|\Psi_0>.

"time-dependent correlation function \tilde{C}_{jl}(t) [cf. (5)] <- This doesn't explain at all how this is defined. From the context it is clear that this refers to <\Psi(t)|c_j c_l^{\dagger}|\Psi(t)>, and therefore there is no good reason why both zero-time and t>0 correlation functions can't be defined with the same equation. Moreover, I do not understand why one of them is denoted by C and the other \tilde{C} - especially since in the next section authors introduce C which is the correlation function in the presence of dissipation.

Line after eq. (11):
"Here, {x,y}=xy+yx is the anticommutator" <- Why is the anticommutator introduced here, if it is already used on the previous page?

Eq. (11-12):
It could be useful to add aditional details on how (12) follows from (11).

Eq. (18):
Where does this follow from? Can a few words be added to support this? I realize it is an old result, but some additional explanation could be useful.

Eq. (20) & line before that:
Maybe the same letter can be used for the subsystem, i.e. instead of W in the text and X in the equation it could both be X or both W.

Sec. 3: Could a few words be added to say whether or not fermionic negativity and usual negativity are equivalent in some sense? How do I know that they measure the same thing?

Line before eq. (25):
"The central object is the fermionic correlation matrix C_{jl} (cf. (5)).". <- Are the authors referring to C of dissipative or non-dissipative case? They are using C as in the dissipative definition, but they are referring to the non-dissipative quantity. I am assuming it's the dissipative one, so reference to eq. (5) is probably wrong.

Line before Eq. (28):
What is C_A? I assume it's C restricted to the subsystem A, but it's never defined.

Eq. (29):
How do we get this expression? A few words of explanation would be nice.

Last paragraph on pg. 8:
"Very recently, Eq. (30) has been verified for quenches in the rule 54 chain [80]." <- This is wrong, Ref. [80] does not deal with Rule 54, but with general circuits (see a remark above).

Last paragraph before Sec. 6:
This is not a "discussion" of the regime of validity of (87), the paragraph is just reiterating a statement from before.

---

## Round 1 · Referee Report · Anonymous (Referee 2) · 2022-8-4

Strengths

1- timely topic

2-interesting quantity considered

Weaknesses

1- presentation focused on the technical aspects and not on the physics of the results

2-poor discussion of the results and comparison with previous literature

Report

The authors have investigated the fermionic logarithmic negativity (obtained from the partial time-reversed reduced density matrix,
not from the standard partial transposition) for a tight binding chain on the line after a a global quantum quench from the Need state.
The system is open, hence dissipation occurs an this is main novelty with respect to the existing literature.
The subsystem is made by the union of two disjoint intervals and the partial time reversal for one of them is considered.
They mainly work in the weakly-dissipative hydrodynamic limit. In this regime, analytic expressions can be found by extending
existing methods developed for the closed systems.

The authors extends to this quantity the analyses they did earlier, in other papers, for the entanglement entropies in a similar setup.
The main result of the manuscript under review is described in sec. 5 (see e.g. eq. (87)) and the corresponding numerical checks are discussed in sec. 6.

Understanding entanglement in open quantum systems is important and timely.
The results of the manuscript are interesting along this line and nicely reported, hence I support the publication of this paper in SciPost.

Requested changes

The major issues to address are

(A) As for the motivations, the authors should motivate why they consider the so-called fermionic negativity and not the standard one
for free fermions. Are there physical reasons or just technical ones? Furthermore,
for the non-experts, the authors should remark the differences between these two quantities.

(B) in the text of pag. 7, three lines below eq. (23) the authors claim that the standard fermionic negativity “cannot be easily calculated, not even for free fermion models”.
Beside the technical difficulties, analytic results have been obtained in the literature (e.g. 1508.00811, 1503.09114, 1601.00678) that the authors could cite

(C) In sec. 4 the authors claim that their previous result for the entanglement entropies is obtained here in a different way.
However, it is not clear which is the novelty of this derivation and why it should be preferred to the previous one.
The derivation is presented as a list of technical steps without any discussion or comparison with the previous analysis.
Which are the steps of the derivation that must be changed with respect to the procedure followed in ref. [81]?
Where the procedure becomes simpler in the limiting regime of closed system?

(D) The main result is presented at pag 19. Just a technical description of the formulas is given.
It is highly recomended to enlarge this discussion by including considerations about the physics of the result.
For instance, a comparison can be made with existing results about the temporal evolution of the negativity,
e.g. against ref. [72] where the typical “rise and fall” behaviour shown in fig. 4 for the negativity has been first obtained for the standard negativity
(for closed systems) or against the temporal evolution of the fermionic one for closed systems.
It could be useful also to have a plot like the the one in fig. 4 in the case of balanced loss/gain dissipation
showing what happens in the limiting regime when $\gamma^+ = \gamma^-$ goes to zero and the result without dissipation is recovered.

(E) an important consequence of the main result is the relation (90), which does not hold in the generic case, as the authors properly remark.
Why a similar relation cannot be found for the generic unbalanced case?
What are the difficulties? Just technically complicated expressions to deal with or more conceptual obstacles occur?

(F) As overall remark, I found the presentation too technical. It seems to me that many technical details could have been appendix material.
During the revision, the authors might consider the possibility to move some derivations in the appendix, in order to focus the attention of the reader
on the result and on its interpretation, and not on the steps of its derivation.

Minor issues and typos are the following

(I) the authors should clarify better the evolution dynamics.
For instance below eq. (3) they claim “non equilibrium dynamics after the quench from the fermionic Neel state” but it is not clear
what is the final stage of the quench.

(II) in eq. (20) the subindex X could be W, to be consistent with the previous text.

(III) in the caption of figs. 4-5-6-7, it would be useful to indicate the equations of the text employed for the analytic curves.

(IV) typo in eq. (56): parenthesis are missing in the l.h.s., in order to be consistent e.g. with eq. (53).

(V) all the full stops/comas at the end of the formulas should be revised
(see e.g. eps. (37), (44), (45), (52), (62), (64), (74), (76) and presumably some other ones)

(VI) in eq. (90) it could be helpful to add the text “ for $gamma^+ = gamma^-$ ” within the equation , as done in eq. (89)

---

## Round 2 · Referee Report · Anonymous · 2023-7-16

Strengths

1. interesting topic
2. some analytic results

Weaknesses

1. derivations of the results poorly described
2. physical implications of the results poorly discussed

Report

Given the previous report, I think that some improvements have been made in the discussion of the main result at pag. 19.

---

## Round 2 · Author Response

Errors in user-supplied markup (flagged; corrections coming soon)

Dear Editor,

We would like to thank you for your work and the referees for their valuable comments.
In the new version of the manuscript we addressed all their criticisms. Below you can find
our reply to their reports.

Sincerely Yours,

Vincenzo Alba
Federico Carollo
* * *
REPLY TO REFEREES
* * *
REFEREE I

############
R: The authors have investigated the fermionic logarithmic negativity (obtained from the partial time-reversed reduced density matrix,
not from the standard partial transposition) for a tight binding chain on the line after a a global quantum quench from the Need state.
The system is open, hence dissipation occurs an this is main novelty with respect to the existing literature.
The subsystem is made by the union of two disjoint intervals and the partial time reversal for one of them is considered.
They mainly work in the weakly-dissipative hydrodynamic limit. In this regime, analytic expressions can be found by extending
existing methods developed for the closed systems.

The authors extends to this quantity the analyses they did earlier, in other papers, for the entanglement entropies in a similar setup.
The main result of the manuscript under review is described in sec. 5 (see e.g. eq. (87)) and the corresponding numerical checks are discussed in sec. 6.

Understanding entanglement in open quantum systems is important and timely.
The results of the manuscript are interesting along this line and nicely reported, hence I support the publication of this paper in SciPost.

A: We thank the referee for considering our work important and timely.

############
R: As for the motivations, the authors should motivate why they consider the so-called fermionic negativity and not the standard one
for free fermions. Are there physical reasons or just technical ones? Furthermore,
for the non-experts, the authors should remark the differences between these two quantities.

A: We regret that in the introduction we did not manage to motivate enough our choice
of the fermionic negativity. As stressed now in the introduction the main reason to
consider the fermionic negativity is its calculability. Indeed, while the computational
cost to calculate the standard negativity is exponential in the system size, the one
to compute the fermionic negativity is only polynomial because the fermionic negativiy
is computed from the two-point correlation function. As for the differences between
the two negativities, we now stress in the introduction that the two quantities
share the same ``good'' properties of entanglement measures for mixed states.
So in conclusion, the both the negativities can be used to quantify
quantum correlations in mixed states. This is also supported by the fact that
for unitary dynamics in closed quantum systems the two negativities become the
same in the scaling limit.

###########
R: in the text of pag. 7, three lines below eq. (23) the authors claim that the standard fermionic negativity
“cannot be easily calculated, not even for free fermion models”. Beside the technical difficulties, analytic
results have been obtained in the literature (e.g. 1508.00811, 1503.09114, 1601.00678) that the authors could cite

A: We thank the referee for pointing to us these papers. We are sorry that we missed them in the first version of
the manuscript. Indeed, these papers are important, especially because they study the moments of the partial
transpose density matrix, and in a way they motivated the introduction of the fermionic logarithmic negativity.
We now mention these important papers in the manuscript.

###########
R: In sec. 4 the authors claim that their previous result for the entanglement entropies is obtained here in a different way.
However, it is not clear which is the novelty of this derivation and why it should be preferred to the previous one.
The derivation is presented as a list of technical steps without any discussion or comparison with the previous analysis.
Which are the steps of the derivation that must be changed with respect to the procedure followed in ref. [81]?
Where the procedure becomes simpler in the limiting regime of closed system?

A: We regret that we were not sufficiently clear about this point in the manuscript. The results presented in sec. 4 can
be considered original because the derivation that we provide is different from that given in our previous
work. In fact, the main virtue of the derivation in sec. 4 is that it is ab initio. In contrast,
the result in Ref. [7] were heavily based on the final formula (61) of Ref. [84]. To apply the result
of Ref. [84] one has to notice that our initial state, i.e., the Neel state, is not translational invariant.
On the other hand, the results of Ref. [84] hold for quenches in the XY chain from generic translation
invariant initial states. This difficulty can be overcome, as discussed in Ref. [7], by transforming the
initial state and the Hamiltonian with a local unitary operator. The problem now becomes that of the dynamics
from a ferromagnetic state (which is translation invariant) with a ``rotated'' Hamiltonian, which is still in the
class of XY models treated in Ref. [84]. Moreover, the rotation is not crucial because the Renyi
entropies are not affected by the unitary transformation. However,
another difficulty is that the results of Ref. [84] hold for
the even moments of the Majorana fermionic correlation matrix, because in the unitary settings the odd
ones are zero. This is not the case in the presence of gain and loss dissipation. To overcome this issue
in Ref. [7] we restricted ourselves to the case of balanced gain and losses. In that case the even moments
of the fermionic correlator are sufficient to obtain the entropies. This allowed us to obtain our conjecture
for the entropies (Eq. (13)) for \gamma^+=\gamma^-. From (13) it was straightforward to conjecture the
result for arbitrary \gamma^+,\gamma^-.

Now, the calculation of the fermionic negativity is much more involved than that of the Renyi entropies and
applying directly the results of Ref. [84] is not as easy. For this reason, and as a warm-up before
attacking the negativity, we decided to rederive the result for the entropies. We believe that the derivation
is instructive, also considering the it takes only few pages and it can be easily followed by the reader
interested in generalizing our results.

This is now better presented in the new version of the manuscript. For instance, now we write explicitly at the
beginning of the section that our derivation is ab initio, although, of course, our approach is similar
to Ref. [84], which pioneered this type of calculations. We also stress some aspects that are different
as compared with the derivation of Ref. [84]. For instance, the fact that the initial state is two-site
translation invariant implies that within the stationary phase approximation the stationary point is not
unique but there is a proliferation of stationary points, due to the invariance by a \pi shift.

###########
R: The main result is presented at pag 19. Just a technical description of the formulas is given.
It is highly recomended to enlarge this discussion by including considerations about the physics of the result.
For instance, a comparison can be made with existing results about the temporal evolution of the negativity,
e.g. against ref. [72] where the typical “rise and fall” behaviour shown in fig. 4 for the negativity has been first obtained for the standard negativity
(for closed systems) or against the temporal evolution of the fermionic one for closed systems.
It could be useful also to have a plot like the one in fig. 4 in the case of balanced loss/gain dissipation
showing what happens in the limiting regime when
\gamma^+=\gamma^- goes to zero and the result without dissipation is recovered.

A: We thank the referee for this important comment. We now extended significantly the discuss of the physical
origin and consequences of our formula. In particular, we now compare the dynamics of negativity in the presence
dissipation with the results of Ref. [72]. In the new paragraph we stress that in the presence of dissipation
the linear growth at short times is damped exponentially although the negativity exhibits the typical ``rise and fall''
dynamics as in the unitary case. We also stress, again, that the negativity content of the quasiparticles is
not the same as the Renyi mutual information. This is in contrast with the unitary case. Following the suggestion
of the referee we also added a new figure for the case of balanced gain and loss dissipation. We also discuss the
structure of the negativity content. In fact we comment on the possibility that the negativity
content of the quasiparticles could be obtained from an effective few-states mixed density matrix for the two subsystems.

###########
R: an important consequence of the main result is the relation (90), which does not hold in the generic case, as the authors properly remark.
Why a similar relation cannot be found for the generic unbalanced case?
What are the difficulties? Just technically complicated expressions to deal with or more conceptual obstacles occur?

A: As we stressed in the previous point it is expected that the negativity content of the quasiparticles
can be obtained from an effective few-states mixed density matrix for the two subsystems. The size of this
density matrix is expected to depend on the number of dissipative processes that can affect the two
subsystems. One could imagine that for independent gain and losses there should be a 4x4 matrix. In the case
of balanced gain and losses one could expected that gain and losses can be treated as a single dissipative
process. This would lead to a reduction of the effective density matrix. This suggests that the fact that
for balanced gain and losses we obtain the simple expression (90) is ``accidental'', and a similar formula
for generic gain and losses is not expected.

###########
R: As overall remark, I found the presentation too technical. It seems to me that many technical details could have been appendix material.
During the revision, the authors might consider the possibility to move some derivations in the appendix, in order to focus the attention of the reader
on the result and on its interpretation, and not on the steps of its derivation.

A: We thank the referee for this suggestion. We moved some of the technical derivations
in the Appendix. We kept the derivation of the moments Tr(G^+G^-)^n in section 5.1.
In section 5.2 we report the main formulas for the moments with defects insertions,
which we believe are interesting. The derivation is reported in Appendix A.

###########
R: the authors should clarify better the evolution dynamics.
For instance below eq. (3) they claim “non equilibrium dynamics after the quench from the fermionic Neel state” but it is not clear
what is the final stage of the quench.

A: We thank the referee for the remark. We now put more details on the quench protocol.

###########
R: in eq. (20) the subindex X could be W, to be consistent with the previous text.

A: We corrected that.

###########
R:in the caption of figs. 4-5-6-7, it would be useful to indicate the equations of the text employed for the analytic curves.

A: We modified the captions of the figures accordingly.

###########
R:typo in eq. (56): parenthesis are missing in the l.h.s., in order to be consistent e.g. with eq. (53).

R: Fixed.

###########
R: all the full stops/comas at the end of the formulas should be revised
(see e.g. eps. (37), (44), (45), (52), (62), (64), (74), (76) and presumably some other ones)

A: We thank the referee for this comment. We fixed the full stops/comas.

###########
R: in eq. (90) it could be helpful to add the text “ for
\gamma^+=\gamma^-” within the equation , as done in eq. (89)

A: We modified the manuscript accordingly.

REFEREE II

###########
R: The authors study the dynamics of logarithmic negativity in an open free-fermion chain. They consider a tight-binding model with the addition of on-site dissipation. After introducing the model, and the so-called "fermionic negativity" (a quantity closely related to negativity that is easier to calculate in free-fermionic systems), they derive expressions describing the dynamics of Renyi entropies and negativity, valid in the special scaling limit of subsystem sizes and time going to infinity with fixed ratios, which they refer to as the "hydrodynamic limit", while the dissipative rates need to scale as the inverse of the subsystem size (referred to as "weakly-dissipative" limit). Interestingly, they find that the usual relation between the Renyi-1/2 mutual information and negativity ceases to hold. Surprisingly, in certain limits they recover an analogous relation between negativity and Renyi-2 mutual information. They also provide numerical data that supports the analytical predictions.

I believe that the main results of the paper are interesting and deserve publication, but not in the current form. Derivations are hard to follow up to the point of the paper being almost unreadable. I am aware that this is a calculation-heavy paper, and is therefore technical in nature. However, the authors should make a stronger effort to make the paper more accessible. I propose the authors try to clean the calculations up, add explanations where relevant, and maybe move some of the more technical parts to an appendix. Moreover, the sections preceding the derivation are missing some more intuitive explanation of the physics discussed there. It feels like a recount of previous results by the authors without enough context to make the paper self contained. I do not expect the authors to rederive all the previous results leading to the calculation in Sec. 5, but it would be nice to get an idea of where most of the statements come from without having to be thouroughly familiar with an extensive list of literature.

Furthermore, I find that the writing is at certain points sloppy with some statements that are misleading or even plainly false. Below I'm attaching a list of examples that I found.

A: We thank the referee for the positive assessment. In the new version of the manuscript we addressed all
the criticisms of the referee.

###########
R: 1st paragraph:
"While for bipartite quantum systems in a pure state several quantum
information motivated measures can be used to identify entanglement [1-4],
this is not the case if the state of the full system is mixed."
<- This seems to be incorrect: the authors in the next paragraph
introduce a measure for mixed-state entanglement, which is also motivated by quantum information.

A: We thank the referee for the comment. We agree that the sentence might sound contradictory. We meant that
while for pure states there are several computable measures of entanglement, such as von Neumann and Renyi
entropies, for mixed states the negativity is the only one that is computable.
Moreover, as we write in the introduction the calculation of the negativity is a
challenging task except for free bosons. Indeed, for free fermions, only the fermionic
negativity can be computed effectively. In numerical simulations with DMRG the algorithms to compute the negativity are
not as effective as the ones for the entropies. Indeed, the typical computational cost to compute the
negativity scales as \chi^6 with \chi the bond dimension. This has to be compared with the cost \propto\chi^3 for the
entropies. Finally, there is no efficient quantum Monte Carlo algorithm to compute the negativity, whereas
the Renyi entropies can be computed effectively with QMC.
We modified the sentence in question as

"While for bipartite quantum systems in a pure state several computable quantum information motivated
measures can be used to identify entanglement, this is more challenging for mixed-state systems"

###########
R: (II) Page 2, 2nd paragraph:
"For interacting integrable systems no results are available." <- This is again
not true. Ref. [80] suggests that in the early-time regime and for
contiguous tripartition this relation between Renyi-1/2 mutual
information and logarithmic negativity holds exactly (i.e. not
only in the scaling limit). This result holds for any local
quantum circuit, and therefore also for interacting integrable
systems. See also Ref. [75] for a similar claim in the case of CFTs.

A: We thank the referee for this comment. Indeed, the paragraph was not precise. We modified
as

For generic interacting systems
it is quite challenging to build a quasiparticle picture to describe the full-time
dynamics of R\'enyi entropies, although
their value in the steady state can be determined.
Moreover, recent exact results for quenches in the so-called rule
$54$ chain~\cite{klobas2021entanglement}, which is a ``minimal model''
for interacting integrable systems, suggest that R\'enyi entropies violate
the quasiparticle picture paradigm. These results motivated a
conjecture for the growth with time of the R\'enyi entropies in
generic interacting integrable systems~\cite{bertini2022growth}.
However, Ref.~\cite{bertini2022quantum} showed that in the early-time regime and for
contiguous subsystems the relation between Renyi-$1/2$ mutual
information and logarithmic negativity put forward in Ref.~\cite{alba2019quantum}
still holds for any local
quantum circuit, and therefore also for interacting integrable
systems. Similar results were obtained in CFTs~\cite{kudler2021the}.

###########
R: "Despite this scenario, it is possible to obtain analytically
the dynamics of the logarithmic negativity in the rule 54 chain,
and it is in agreement with Ref. [23]." <- Again, this is misleading.
As far as I'm aware there is no explicit calculation for Rule 54.
The only result that could come close to that is the general
statement of Ref. [80] that holds for *all* local unitary
circuits (see above), but this is only in agreement with
Ref. [23] in the early-time regime.

A: We modified the text to be more precise, see previous point.

###########
R: Sec. 2: I am assuming that L is even. It could be useful to point it out explicitly.

A: We thank the referee for the comment. We now write that L is even below Eq. (4).

###########
R: Eq. (5): It could be useful to define what kind of expectation value it is, i.e.
C_{jl}:=<c_j c_l^{\dagger}>=<\Psi_0|c_j^{\dagger}c_l|\Psi_0>.

A: In the new version of the manuscript we implemented the suggestion of the referee.

###########
R: "time-dependent correlation function \tilde{C}_{jl}(t) [cf. (5)] <-
This doesn't explain at all how this is defined. From the context it
is clear that this refers to <\Psi(t)|c_j c_l^{\dagger}|\Psi(t)>, and therefore
there is no good reason why both zero-time and t>0 correlation
functions can't be defined with the same equation. Moreover, I
do not understand why one of them is denoted by C and the other
\tilde{C} - especially since in the next section authors
introduce C which is the correlation function in the presence of dissipation.

A: We thank the referee for this comment. The reason of the confusion is that we
had in mind of defining with the tilde the correlation function for the case
without dissipation and without the tilde the correlation in the presence of both
dissipative and unitary dynamics. To make this clear we now use the tilde in section 2 and
the notation without the tilde in 2.1.

###########
R: Line after eq. (11):
"Here, {x,y}=xy+yx is the anticommutator" <- Why is the anticommutator
introduced here, if it is already used on the previous page?

A: We removed the sentence in question.

###########
R: Eq. (11-12): It could be useful to add aditional details on how (12) follows from (11).

A: We thank the referee for the comment. We now added a sentence to make explicit that (12) is
obtained from (11) just using the definition of the fermionic correlation function,
the dynamics of the density matrix, and Wick's theorem to perform the trace. We also add a
Ref. [33] where the formula is reported and further explained.

###########
R: Eq. (18):
Where does this follow from? Can a few words be added to support this? I realize it is an old result, but some additional explanation could be useful.

A: The referee is indeed right that as it is the sentence is not very clear. We added few sentences to
explain how (18) can be derived. The key point is that in the limit of small dissipation rates, (18) is to
be expected if one assumes that the eigenvalue and the eigenvector of the Liouvillian differ from those of the
Hamiltonian by terms that are linear in the dissipation rates.

###########
R: Eq. (20) & line before that:
Maybe the same letter can be used for the subsystem, i.e. instead of W in the text and X in the
equation it could both be X or both W.

A: We corrected according to the referee suggestion.

###########
R: Sec. 3: Could a few words be added to say whether or not fermionic negativity and usual negativity are equivalent in some sense? How do I know that they measure the same thing?

A: We thank the referee for the comment. First, the fermionic and the standard negativity do not coincide.
However, they both can be taken as bona fide entanglement measure, as we now write explicitly in the introduction.
The meaning of this fact is discussed in Ref. [21].

###########
R: Line before eq. (25):
"The central object is the fermionic correlation matrix C_{jl} (cf. (5)).". <- Are the authors referring to C of dissipative or non-dissipative case? They are using C as in the dissipative definition, but they are referring to the non-dissipative quantity. I am assuming it's the dissipative one, so reference to eq. (5) is probably wrong.

A: We modified the notation. Now the correlator for the case without dissipation is denoted by a tilde.
We now refer to (12) for the correlator that is needed to construct the negativity.

###########
R: Line before Eq. (28):
What is C_A? I assume it's C restricted to the subsystem A, but it's never defined.

A: The referee is right that C_A is the restricted correlation matrix. This is now
explicitly stated.

###########
R:Eq. (29):
How do we get this expression? A few words of explanation would be nice.

A: In the new version of the manuscript we added a few remarks on the meaning of the
two terms in (29). We also cite a new reference [51] where that formula is derived.

###########
R: Last paragraph on pg. 8:
"Very recently, Eq. (30) has been verified for quenches in the rule 54 chain [80]." <- This is wrong, Ref. [80] does not deal with Rule 54, but with general circuits (see a remark above).

A: We thank the referee. We corrected the sentence.

###########
R: Last paragraph before Sec. 6:
This is not a "discussion" of the regime of validity of (87), the paragraph is just reiterating a statement from before.

A: We agree with the referee that this is not a discussion. However, it is not just a reiteration of a statement
that is present in section 5.3. Thus, we prefer to keep it to make it clear the regime of validity of
the result. We rephrased the final paragraph to make it clear that it not as a discussion.

---

## Editorial Decision

published